# Can I Trust My Fairness Metric? Assessing Fairness with Unlabeled Data and Bayesian Inference

**Disi Ji**[1]    **Padhraic Smyth**[1]    **Mark Steyvers**[2]

[1]Department of Computer Science    [2]Department of Cognitive Sciences

University of California, Irvine

disij@uci.edu    smyth@ics.uci.edu    mark.steyvers@uci.edu

## Abstract

We investigate the problem of reliably assessing group fairness when labeled examples are few but unlabeled examples are plentiful. We propose a general Bayesian framework that can augment labeled data with unlabeled data to produce more accurate and lower-variance estimates compared to methods based on labeled data alone. Our approach estimates calibrated scores for unlabeled examples in each group using a hierarchical latent variable model conditioned on labeled examples. This in turn allows for inference of posterior distributions with associated notions of uncertainty for a variety of group fairness metrics. We demonstrate that our approach leads to significant and consistent reductions in estimation error across multiple well-known fairness datasets, sensitive attributes, and predictive models. The results show the benefits of using both unlabeled data and Bayesian inference in terms of assessing whether a prediction model is fair or not.

## 1   Introduction

Machine learning models are increasingly used to make important decisions about individuals. At the same time it has become apparent that these models are susceptible to producing systematically biased decisions with respect to sensitive attributes such as gender, ethnicity, and age [Angwin et al., 2017, Berk et al., 2018, Corbett-Davies and Goel, 2018, Chen et al., 2019, Beutel et al., 2019]. This has led to a significant amount of recent work in machine learning addressing these issues, including research on both (i) definitions of fairness in a machine learning context (e.g., Dwork et al. [2012], Chouldechova [2017]), and (ii) design of fairness-aware learning algorithms that can mitigate issues such as algorithmic bias (e.g., Calders and Verwer [2010], Kamishima et al. [2012], Feldman et al. [2015], Zafar et al. [2017], Chzhen et al. [2019]).

In this paper we focus on an important yet under-studied aspect of the fairness problem: reliably assessing how fair a blackbox model is, given limited labeled data. In particular, we focus on assessment of group fairness of binary classifiers. Group fairness is measured with respect to parity in prediction performance between different demographic groups. Examples include differences in performance for metrics such as true positive rates and false positive rates (also known as equalized odds [Hardt et al., 2016]), accuracy [Chouldechova, 2017], false discovery/omission rates [Zafar et al., 2017], and calibration and balance [Kleinberg et al., 2016].

Despite the simplicity of these definitions, a significant challenge in assessment of group fairness is high variance in estimates of these metrics based on small amounts of labeled data. To illustrate this point, Figure 1 shows frequency-based estimates of group differences in true positive rates (TPRs) for four real-world datasets. The boxplots clearly show the high variability for the estimated TPRs relative to the true TPRs (shown in red) as a function of the number of labeled examples $n_L$. In many cases the estimates are two or three or more times larger than the true difference. In addition, a relatively large percentage of the estimates have the opposite sign of the true difference, potentially

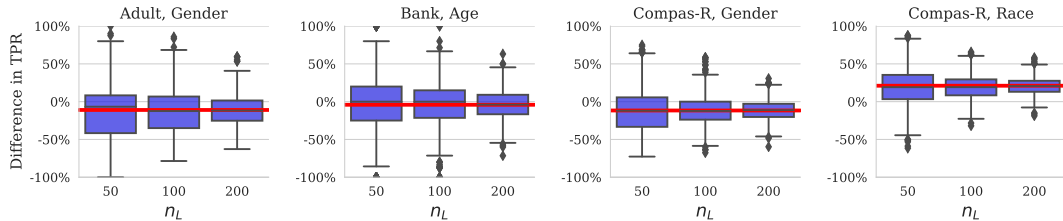

Figure 1: Boxplots of frequency-based estimates of the difference in true positive rate (TPR) for four fairness datasets and binary sensitive attributes, across 1000 randomly sampled sets of labeled test examples of size $n_L = 50, 100, 200$. The horizontal red line is the TPR difference computed on the full test dataset.

leading to mistaken conclusions. The variance of these estimates decreases relatively slowly, e.g., at a rate of approximately $\frac{1}{n}$ for group differences in accuracy where $n$ is the number of labels in the smaller of the two groups[1]. Imbalances in label distributions can further compound the problem, for example for estimation of group differences in TPR or FPR. For example, consider a simple simulation with two groups, where the underrepresented group makes up 20% of the whole dataset, groupwise positive rates $P(y = 1)$ are both 20%, and the true groupwise TPRs are 95% and 90% (additional details in the Supplement). In order to ensure that there is a 95% chance that our estimate of the true TPR difference (which is 0.05) lies in the range [0.04, 0.06] we need at least 96k labeled instances. Yet for real-world datasets used in the fairness literature (e.g., Friedler et al. [2019]; see also Table 1 later in the paper), test set sizes are often much smaller than this, and it is not uncommon for the group and label distributions to be even more imbalanced.

The real-world and synthetic examples above show that frequentist assessment of group fairness is unreliable unless the labeled dataset is unrealistically large. Acquiring large amounts of labeled data can be difficult and time-consuming, particularly for the types of applications where fairness is important, such as decision-making in medical or criminal justice contexts [Angwin et al., 2017, Berk et al., 2018, Rajkomar et al., 2018]. This is in contrast to applications such as image classification where approaches like Mechanical Turk can be used to readily generate large amounts of labeled data.

To address this problem, we propose to augment labeled data with unlabeled data to generate more accurate and lower-variance estimates compared to methods based on labeled data alone. In particular, the three primary contributions of this paper are (1) a comprehensive Bayesian treatment of fairness assessment that provides uncertainty about estimates of group fairness metrics; (2) a new Bayesian methodology that uses calibration to leverage information from both unlabeled and labeled examples; and, (3) systematic large-scale experiments across multiple datasets, models, and attributes that show that using unlabeled data can reduce estimation error significantly.

## 2 Fairness Assessment with Bayesian Inference and Unlabeled Data

### 2.1 Notation and Problem Statement

Consider a trained binary classification model $M$, with inputs $x$ and class labels $y \in \{0, 1\}$. The model produces scores[2] $s = P_M(y = 1|x) \in [0, 1]$, where $P_M$ denotes the fact that this is the model's estimate of the probability that $y = 1$ conditioned on $x$. Under 0-1 loss the model predicts $\hat{y} = 1$ if $s \geq 0.5$ and $\hat{y} = 0$ otherwise. The marginal accuracy of the classifier is $P(\hat{y} = y)$ and the accuracy conditioned on a particular value of the score $s$ is $P(\hat{y} = y|s)$. A classifier is calibrated if $P(\hat{y} = y)|s) = s$, e.g., if whenever the model produces a score of $s = 0.9$ then its prediction is correct 90% of the time. For group fairness we are interested in potential differences in performance metrics with respect to a sensitive attribute (such as gender or race) whose values $g$ correspond to different groups, $g \in \{0, 1, \ldots, G - 1\}$. We will use $\theta_g$ to denote a particular metric of interest, such

as accuracy, TPR, FPR, etc. for group $g$. We focus on group differences for two groups, defined as $\Delta = \theta_1 - \theta_0$, e.g., the difference in a model's predictive accuracy between females and males, $\Delta = P(\hat{y} = y|g = 1) - P(\hat{y} = y|g = 0)$.

We assume in general that the available data consists of both $n_L$ labeled examples and $n_U$ unlabeled examples, where $n_L \ll n_U$, which is a common situation in practice where far more unlabeled data is available than labeled data. For the unlabeled examples, we do not have access to the true labels $y_j$ but we do have the scores $s_j = P_M(y_j = 1|x_j)$, $j = 1, \ldots, n_U$. For the labeled examples, we have the true labels $y_i$ as well as the scores $s_i$, $i = 1, \ldots, n_L$. The examples (inputs $x$, scores $s$, and labels $y$ if available) are sampled IID from an underlying joint distribution $P(x, y)$ (or equivalently $P(s, y)$ given that $s$ is a deterministic function via $M$ of $x$), where this underlying distribution represents the population we wish to evaluate fairness with respect to. Note that in practice $P(x, y)$ might very well not be the same distribution the model was trained on. For unlabeled data $D_u$ the corresponding distributions are $P(x)$ or $P(s)$.

## 2.2   Beta-Binomial Estimation with Labeled Data

Consider initially the case with only labeled data $D_L$ (i.e., $n_U = 0$) and for simplicity let the metric of interest $\Delta$ be group difference in classification accuracy. Let $I_i = I_{\hat{y}_i = y_i}, 1 \leq i \leq n_L$, be a binary indicator of whether each labeled example $i$ was classified correctly or not by the model. The binomial likelihood for group accuracy $\theta_g, g = 0, 1$, treats the $I_i$'s as conditionally independent draws from a true unknown accuracy $\theta_g$, $I_i \sim \mathrm{Bernoulli}(\theta_g)$. We can perform Bayesian inference on the $\theta_g$'s by specifying conjugate $\mathrm{Beta}(\alpha_g, \beta_g)$ priors for each $\theta_g$, combining these priors with the binomial likelihoods, and obtaining posterior densities in the form of the beta densities on each $\theta_g$. From here we can get a posterior density on the group difference in accuracy, $P(\Delta|D_L)$ where $\Delta = \theta_1 - \theta_0$. Since the density for the difference of two beta-distributed quantities (the $\theta$'s) is not in general in closed form, we use posterior simulation (e.g., Gelman et al. [2013]) to obtain posterior samples of $\Delta$ by sampling $\theta$'s from their posterior densities and taking the difference. For metrics such as TPR we place beta priors on conditional quantities such as $\theta_g = P(\hat{y} = 1|y = 1, g)$. In all of the results in the paper we use weak uninformative priors for $\theta_g$ with $\alpha_g = \beta_g = 1$. This general idea of using Bayesian inference on classifier-related metrics has been noted before for metrics such marginal accuracy [Benavoli et al., 2017], TPR [Johnson et al., 2019], and precision-recall [Goutte and Gaussier, 2005], but has not been developed or evaluated in the context of fairness assessment.

This beta-binomial approach above provides a useful, simple, and practical tool for understanding and visualizing uncertainty about fairness-related metrics, conditioned on a set of $n_L$ labeled examples. However, with weak uninformative priors, the posterior density for $\Delta$ will be relatively wide unless $n_L$ is very large, analogous to the high empirical variance for frequentist point estimates in Figure 1. As with the frequentist variance, the width of the posterior density on $\Delta$ will decrease relatively slowly at a rate of approximately $\frac{1}{n_L}$. This motivates the main goal of the paper: can we combine unlabeled examples with labeled examples to make more accurate inferences about fairness metrics?

## 2.3   Leveraging Unlabeled Data with a Bayesian Calibration Model

Consider the situation where we have $n_U$ unlabeled examples, in addition to the $n_L$ labeled ones. For each unlabeled example $j = 1, \ldots, n_U$ we can use the model $M$ to generate a score, $s_j = P_M(y_j = 1|x_j)$. If the model $M$ is perfectly calibrated then the model's score is the true probability that $y = 1$, i.e., we have $s_j = P_M(y_j = 1|s_j)$ and the accuracy equals $s_j$ if $s_j \geq 0.5$ and $1 - s_j$ otherwise. Therefore, in the perfectly calibrated case, we could empirically estimate accuracy per group for the unlabeled data using scores via $\hat{\theta}_g = (1/n_{U,g}) \sum_{j \in g} s_j I(s_j \geq 0.5) + (1 - s_j) I(s_j < 0.5)$, where $n_{U,g}$ is the number of unlabeled examples that belong to group $g$. Metrics other than accuracy could also be estimated per group in a similar fashion.

In practice, however, many classification models, particularly complex ones such as deep learning models, can be significantly miscalibrated (see, e.g., Guo et al. [2017], Kull et al. [2017], Kumar et al. [2019], Ovadia et al. [2019]) and using the uncalibrated scores in such situations will lead to biased estimates of the true accuracy per group. The key idea of our approach is to use the labeled data to learn how to calibrate the scores such that the calibrated scores can contribute to less biased estimates of accuracy. Let $z_j = E[I(\hat{y}_j = y_j)] = P(y_j = \hat{y}_j|s_j)$ be the true (unknown) accuracy of the model

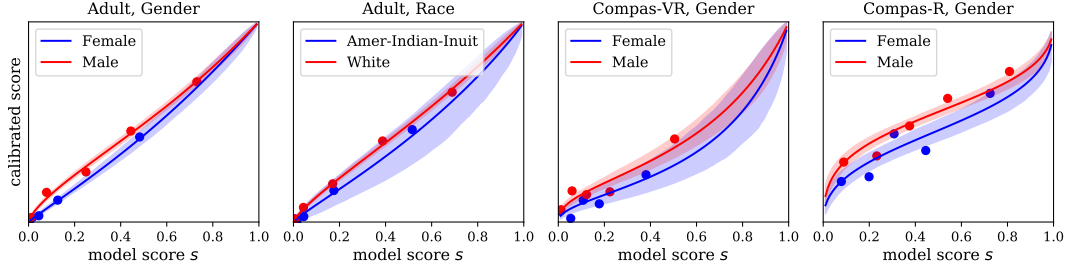

Figure 2: Hierarchical Bayesian calibration of two demographic groups across four dataset-group pairs, with posterior means and 95% credible intervals per group. The $x$-axis is the model score $s$ for class $y = 1$, and the $y$-axis is the calibrated score. Instances in each group are binned into 5 equal-sized bins by model score, and blue and red points show the fraction of positive samples per group for each bin.

given score $s_j$. We treat each $z_j, j = 1, \ldots, n_U$ as a latent variable per example. The high-level steps of the approach are as follows:

- We use the $n_L$ labeled examples to estimate groupwise calibration functions with parameters $\phi_g$, that transform the (potentially) uncalibrated scores $s$ of the model to calibrated scores. More specifically, we perform Bayesian inference (see Section 2.4 below) to obtain posterior samples from $P(\phi_g|D_L)$ for the groupwise calibration parameters $\phi_g$.

- We then obtain posterior samples from $P_{\phi_g}(z_j|D_L, s_j)$ for each unlabeled example $j = 1, \ldots, n_U$, conditioned on posterior samples of the $\phi_g$'s.

- Finally, we use posterior samples from the $z_j$'s, combined with the labeled data, to generate estimates of the groupwise metrics $\theta_g$ and the difference in metrics $\Delta$.

We can compute a posterior sample for $\theta_g^t$, given each set of posterior samples for $\phi_g^t$ and $z_1^t, \ldots, z_{n_U}^t$, by combining estimates of accuracies for the unlabeled examples with the observed outcomes for the labeled instances:

$$\theta_g^t = \frac{1}{n_{L,g} + n_{U,g}} \left( \sum_{i:i \in g} I(\hat{y}_i = y_i) + \sum_{j:j \in g} z_j^t \right) \tag{1}$$

where $t = 1, ..., T$ is an index over $T$ MCMC samples. These posterior samples in turn can be used to generate an empirical posterior distribution $\{\Delta^1, \ldots, \Delta^T\}$ for $\Delta$, where $\Delta^t = \theta_1^t - \theta_0^t$. Mean posterior estimates can be obtained by averaging over samples, i.e. $\hat{\Delta} = (1/T) \sum_t^T \Delta^t$. Even with very small amounts of labeled data (e.g., $n_L = 10$) we will demonstrate later in the paper that we can make much more accurate inferences about fairness metrics via this Bayesian calibration approach, compared to using only the labeled data directly.

## 2.4 Hierarchical Bayesian Calibration

Bayesian calibration is a key step in our approach above. We describe Bayesian inference below for the beta calibration model specifically [Kull et al., 2017] but other calibration models could also be used. The beta calibration model maps a score from a binary classifier with scores $s = P_M(y = 1|x) \in [0, 1]$ to a recalibrated score according to:

$$f(s; a, b, c) = \frac{1}{1 + e^{-c - a \log s + b \log(1-s)}} \tag{2}$$

where $a$, $b$, and $c$ are calibration parameters with $a, b \geq 0$. This model can capture a wide variety of miscalibration patterns, producing the identity mapping if $s$ is already calibrated when $a = b = 1, c = 0$. Special cases of this model are the linear-log-odds (LLO) calibration model [Turner et al., 2014] when $a = b$, and temperature scaling [Guo et al., 2017] when $a = b, c = 0$.

In our hierarchical Bayesian extension of the beta calibration model, we assume that each group (e.g., female, male) is associated with its own set of calibration parameters $\phi_g = \{a_g, b_g, c_g\}$ and therefore

each group can be miscalibrated in different ways (e.g., see Figure 2). To apply this model to the observed data, we assume that the true labels for the observed instances are sampled according to:

$$y_i \sim \text{Bernoulli}\big(f(s_i; a_{g_i}, b_{g_i}, c_{g_i})\big)$$

where $g_i$ is the group associated with instance $i$, $1 \leq i \leq n_L$. For any unlabeled example $j = 1, \ldots, n_U$, conditioned on calibration parameters $\phi_{g_j}$ for the group for $j$, we can compute the latent variable $z_j = f(s_j; \ldots)I(s_j \geq 0.5) + (1 - f(s_j; \ldots))I(s_j < 0.5)$, i.e., the calibrated probability that the model's prediction on instance $j$ is correct.

We assume that the parameters from each individual group are sampled from a shared distribution: $\log a_g \sim \text{N}(\mu_a, \sigma_a), \log b_g \sim \text{N}(\mu_b, \sigma_b), c_g \sim \text{N}(\mu_c, \sigma_c)$ where $\pi = \{\mu_a, \sigma_a, \mu_b, \sigma_b, \mu_c, \sigma_c\}$ is the set of hyperparameters of the shared distributions. We complete the hierarchical model by placing the following priors on the hyperparameters (TN is the truncated normal distribution):

$$\mu_a \sim \text{N}(0, .4), \mu_b \sim \text{N}(0, .4), \mu_c \sim \text{N}(0, 2), \sigma_a \sim \text{TN}(0, .15), \sigma_b \sim \text{TN}(0, .15), \sigma_c \sim \text{TN}(0, .75)$$

These priors were chosen to place reasonable bounds on the calibration parameters and allow for diverse patterns of miscalibration (e.g., both over and under-confidence or a model) to be expressed a priori. We use exactly these same prior settings in all our experiments across all datasets, all groups, and all labeled and unlabeled dataset sizes, demonstrating the robustness of these settings across a wide variety of contexts. In addition, the Supplement contains results of a sensitivity analysis for the variance parameters in the prior, illustrating robustness across a broad range of settings of these parameters.

The model was implemented as a graphical model (see Supplement) in JAGS, a common tool for Bayesian inference with Markov chain Monte Carlo [Plummer, 2003]. All of the results in this paper are based on 4 chains, with 1500 burn-in iterations and 200 samples per chain, resulting in $T = 800$ sample overall. These hyperparameters were determined based on a few simulation runs across datasets, checking visually for lack of auto-correlation, with convergence assessed using the standard measure of within-to-between-chain variability. Although MCMC can sometimes be slow for high-dimensional problems, with 100 labeled data points (for example) and 10k unlabeled data points the sampling procedure takes about 30 seconds (using non-optimized Python/JAGS code on a standard desktop computer) demonstrating the practicality of this procedure.

**Theoretical Considerations:**  Lemma 2.1 below relates potential error in the calibration mapping (e.g., due to misspecification of the parametric form of the calibration function $f(s; \ldots)$) to error in the estimate of $\Delta$ itself.

**Lemma 2.1.** *Given a prediction model $M$ and score distribution $P(s)$, let $f_g(s; \phi_g) : [0, 1] \to [0, 1]$ denote the calibration model for group $g$; let $f_g^*(s) : [0, 1] \to [0, 1]$ be the optimal calibration function which maps $s = P_M(\hat{y} = 1|g)$ to $P(y = 1|g)$; and $\Delta^*$ is the true value of the metric. Then the absolute error of the expected estimate w.r.t. $\phi$ can be bounded as: $|\mathbb{E}_\phi \Delta - \Delta^*| \leq \|\bar{f}_0 - f_0^*\|_1 + \|\bar{f}_1 - f_1^*\|_1$, where $\bar{f}_g(s) = \mathbb{E}_{\phi_g} f_g(s; \phi_g), \forall s \in [0, 1]$, and $\| \cdot \|_1$ is the expected L1 distance w.r.t. $P(s|g)$. (Proof provided in the Supplement).*

Thus, reductions in the L1 calibration error directly reduce an upper bound on the L1 error in estimating $\Delta$. The results in Figure 2 suggest that even with the relatively simple parametric beta calibration method, the error in calibration (difference between the fitted calibration functions) (blue and red curves) and the empirical data (blue and red dots) is quite low across all 4 datasets. The possibility of using more flexible calibration functions is an interesting direction for future work.

## 3   Datasets, Classification Models, and Illustrative Results

One of the main goals of our experiments is to assess the accuracy of different estimation methods, using relatively limited amounts of labeled data, relative to the true value of the metric. By "true value" we mean the value we could measure on an infinitely large test sample. Since such a sample is not available, we use as a proxy the value of metric computed on all of the test set for each dataset in our experiments.

We followed the experimental methods and used the code for preprocessing and training prediction models from Friedler et al. [2019] who systematically compared fairness metrics and fairness-aware

Table 1: Datasets used in the paper. $G$ is the sensitive attribute, $P(g = 0)$ is the probability of the privileged group, and $P(y = 1)$ is the probability of the positive label for classification. The privileged groups $g = 0$ are gender: male, age: senior or adult, and race: white or Caucasian.

| Dataset | Test Size | $G$ | $P(g = 0)$ | $P(y = 1)$ |
|---------|-----------|-----|------------|------------|
| Adult | 10054 | gender, race | 0.68, 0.86 | 0.25 |
| Bank | 13730 | age | 0.45 | 0.11 |
| German | 334 | age, gender | 0.79, 0.37 | 0.17 |
| Compas-R | 2056 | gender, race | 0.7, 0.85 | 0.69 |
| Compas-VR | 1337 | gender, race | 0.8, 0.34 | 0.47 |
| Ricci | 40 | race | 0.65 | 0.50 |

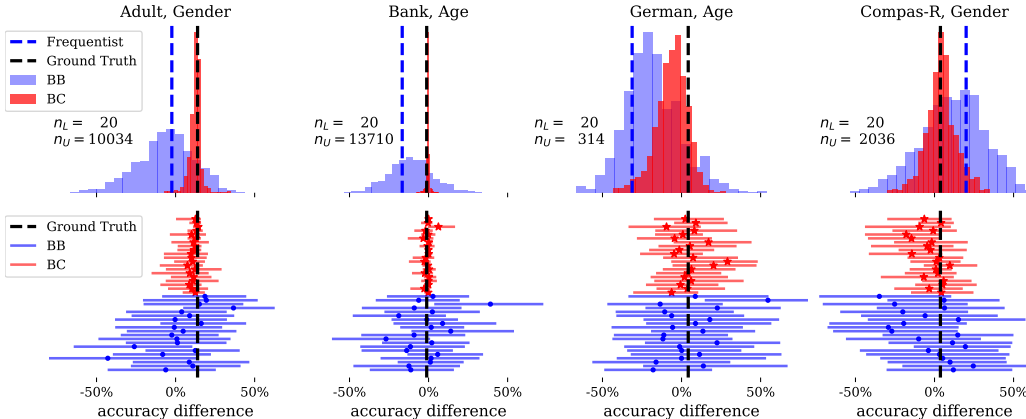

Figure 3: Posterior density (samples) and frequentist estimates (dotted vertical blue lines) for the difference in group accuracy $\Delta$ for 4 datasets with $n_L = 20$ random labeled examples for both the BB (beta-binomial) and BC (Bayesian calibration) methods. Ground truth is a vertical black line. The underlying model is an MLP. The 20 examples were randomly sampled 20 different times. Upper plots show the histograms of posterior samples for the first sample, lower plots show the 95% posterior credible intervals for all 20 runs, where the x-axis is $\Delta$.

algorithms across a variety of datasets. Specifically, we use the Adult, German Credit, Ricci, and Compas datasets (for recidivism and violent recidivism), all used in Friedler et al. [2019], as well as the Bank Telemarketing dataset [Moro et al., 2014]. Summary statistics for the datasets are shown in Table 1. Classification models (logistic regression, multilayer perceptron (MLP) with a single hidden layer, random forests, Gaussian Naive Bayes) were trained using standard default parameter settings with the code provided by Friedler et al. [2019] and predictions generated on the test data. Sensitive attributes are not included as inputs to the models. Unless a specific train/test split is provided in the original dataset, we randomly sample 2/3 of the instances for training and 1/3 for test. Additional details on models and datasets are provided in the Supplement.

**Illustrative Results:** To illustrate our approach we compare the results of the frequentist, beta-binomial (BB), and Bayesian calibration (BC) approaches for assessing group differences in accuracy across 4 datasets, for a multilayer perceptron (MLP) binary classifier. We ran the methods on 20 runs of randomly sampled sets of $n_L = 20$ labeled examples. The BC method was given access to the remaining $n_U$ unlabeled test examples minus the 20 labeled examples for each run, as described in Table 1. We define ground truth as the frequentist $\Delta$ value computed on all the labeled data in the test set. Figure 3 shows the results across the 4 datasets. The top figure corresponds to the first run out of 20 runs, showing the histogram of 800 posterior samples from the BB (blue) and BC (red) methods. The lower row of plots summarizes the results for all 20 runs, showing the 95% posterior credible intervals (CIs) (red and blue horizontal lines for BC and BB respectively) along with posterior means (red and blue marks).

Because of the relatively weak prior (Beta(1,1) on group accuracy) the posterior means of the BB samples tend to be relatively close to the frequentist estimate (light and dark blue respectively) on each run and both can be relatively far away from ground truth value for $\Delta$ (in black). Although

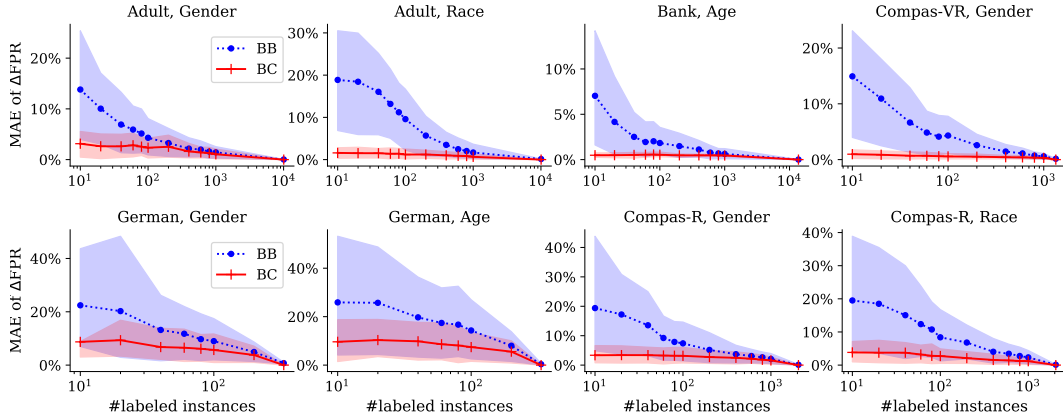

Figure 4: Mean absolute error (MAE) of the difference between algorithm estimates and ground truth for group difference in FPR, as a function of number of labeled instances, for 8 different dataset-group pairs. Shading indicates 95% error bars for each method.

the BB method is an improvement over being frequentist, in that it provides posterior uncertainty about $\Delta$, it nonetheless has high variance (locations of the posterior means) as well as high posterior uncertainty (relatively wide CIs). The BC method in contrast, by using the unlabeled data in addition to the labeled data, produces posterior estimates where the mean tends to be much closer to ground truth than BC.

The posterior information about $\Delta$ can be used to provide users with a summary report that includes information about the direction of potential bias (e.g., $P(\Delta > 0|D_L, D_U)$, the degree of bias (e.g., via the MPE $\hat{\Delta}$), 95% posterior CIs on $\Delta$, and the probability that the model is "practically fair" (assessed via $P(|\Delta| < \epsilon|D_L, D_U)$, e.g., see Benavoli et al. [2017]). For example with BC, given the observed data, practitioners can conclude from the information in the upper row of Figure 3, and with $\epsilon = 0.02$, that there is a 0.99 probability for the Adult data that the classifier is more accurate for females than males; and with probability 0.87 that the classifier is practically fair with respect to accuracy for junior and senior individuals in the Bank data.

## 4 Experiments and Results

In this section we systematically evaluate the quality of different estimation approaches by repeating the same type of experiment as in Section 3 and Figure 3 across different amounts of labeled data $n_L$. In particular, for each value of $n_L$ we randomly sample sets of labeled datasets of size $n_L$, generate point estimates of a metric $\Delta$ of interest for each labeled dataset for each of the BB and BC estimation methods, and compute the mean absolute error (MAE) between the point estimates and the true value (computed on the full labeled test set). The frequency-based estimates are not shown for clarity—they are almost always worse than both BB and BC. As an example, Figure 4 illustrates the quality of estimation where $\Delta$ is the FPR group difference $\Delta$ for the MLP classification model, evaluated across 8 different dataset-group pairs. Each y-value is the average of 100 different randomly sampled sets of $n_L$ instances, where $n_L$ is the corresponding x-axis value. The BC method dominates BB across all datasets indicating that the calibrated scores are very effective at improving the accuracy in estimating group FPR. This is particularly true for small amounts of labeled data (e.g., up to $n_L = 100$) where the BB Method can be highly inaccurate, e.g., MAEs on the order of 10 or 20% when the true value of $\Delta$ is often less than 10%.

In the Supplement we show that the trend of results shown in Figure 4, namely that BC produces significantly more accurate estimates of group fairness metrics $\Delta$, is replicated across all 4 classification models that we investigated, across FPR, TPR and Accuracy metrics, and across all datasets. To summarize the full set of results we show a subset in tabular form, across all 4 classification models and 10 dataset-group pairs, with $n_L$ fixed: Table 2 for Accuracy with $n_L = 10$ and Table 3 for TPR with $n_L = 200$. (We used larger $n_L$ values for TPR and FPR than for accuracy in the results above since TPR and FPR depend on estimating conditional probabilities that can have zero supporting counts in the labeled data, causing a problem for frequentist estimators). The results above and in the

Table 2: **MAE for $\Delta$ Accuracy Estimates**, with $n_L = 10$, across 100 runs of labeled samples, for 4 different trained models (groups of columns) and 10 different dataset-group combinations (rows). Lowest error rate per row-col group in bold if the difference among methods are statistically significant under Wilcoxon signed-rank test (p=0.05). Estimation methods are Freq (Frequentist), BB, and BC. Freq and BB use only labeled samples, BC uses both labeled samples and unlabeled data. Trained models are Multilayer Perceptron, Logistic Regression, Random Forests, and Gaussian Naive Bayes.

| Dataset, Attribute | Multi-layer Perceptron | | | Logistic Regression | | | Random Forest | | | Gaussian Naive Bayes | | |
|---|---|---|---|---|---|---|---|---|---|---|---|---|
| | Freq | BB | BC | Freq | BB | BC | Freq | BB | BC | Freq | BB | BC |
| Adult, Race | 16.5 | 18.5 | **3.9** | 16.4 | 18.7 | **2.9** | 16.5 | 18.2 | **3.2** | 17.6 | 18.9 | **3.6** |
| Adult, Gender | 19.7 | 17.4 | **5.1** | 19.1 | 16.1 | **2.2** | 17.7 | 17.4 | **4.8** | 19.7 | 16.2 | **5.4** |
| Bank, Age | 15.9 | 13.9 | **2.5** | 13.9 | 13.0 | **1.4** | 11.8 | 11.1 | **1.0** | 15.5 | 13.7 | **1.7** |
| German, Age | 34.6 | 19.8 | **5.0** | 37.1 | 21.2 | **8.7** | 33.6 | 18.7 | **8.2** | 36.6 | 20.4 | **11.5** |
| German, Gender | 30.7 | 21.6 | **8.2** | 25.6 | 17.4 | **6.3** | 27.7 | 19.3 | **8.6** | 30.0 | 20.1 | **6.5** |
| Compas-R, Race | 31.5 | 21.0 | **4.2** | 31.7 | 20.4 | **4.8** | 29.3 | 20.3 | **2.4** | 33.5 | 23.2 | **8.4** |
| Compas-R, Gender | 33.7 | 21.6 | **5.0** | 34.3 | 21.9 | **3.8** | 36.3 | 23.3 | **4.4** | 40.5 | 25.5 | **13.7** |
| Compas-VR, Race | 18.7 | 17.1 | **4.0** | 18.5 | 15.6 | **4.4** | 18.2 | 15.8 | **2.4** | 26.6 | 19.8 | **6.5** |
| Compas-VR, Gender | 20.6 | 16.9 | **5.4** | 19.9 | 16.6 | **5.3** | 22.3 | 19.0 | **6.3** | 31.3 | 21.5 | **9.8** |
| Ricci, Race | 23.5 | 17.7 | **14.6** | 14.6 | 14.6 | **7.9** | 6.3 | 12.2 | **2.1** | 8.9 | 13.1 | **1.6** |

Table 3: **MAE for $\Delta$ TPR Estimates**, with $n_L = 200$. Same setup as for Table 2. Compas-VR race and Ricci race are not included since there are no positive instances for some groups, and some entries under Freq cannot be estimated for the same reason.

| Dataset, Attribute | Multi-layer Perceptron | | | Logistic Regression | | | Random Forest | | | Gaussian Naive Bayes | | |
|---|---|---|---|---|---|---|---|---|---|---|---|---|
| | Freq | BB | BC | Freq | BB | BC | Freq | BB | BC | Freq | BB | BC |
| Adult, Race | — | 12.5 | **5.8** | — | 14.7 | **7.0** | — | 14.3 | **4.6** | — | 14.6 | **3.0** |
| Adult, Gender | 16.3 | 14.3 | **4.3** | 15.8 | 14.0 | **4.6** | 16.1 | 14.2 | **7.3** | 15.0 | 13.4 | 11.5 |
| Bank, Age | 16.8 | 15.0 | **4.8** | 17.7 | 15.9 | **4.2** | 16.6 | 14.9 | **3.1** | 17.3 | 15.7 | **2.3** |
| German, Age | 4.7 | 4.7 | **3.0** | 5.6 | 5.4 | **2.6** | 5.1 | 5.1 | **3.1** | 6.8 | 6.5 | **2.8** |
| German, Gender | **0.7** | 1.0 | 1.6 | 3.3 | 3.3 | **2.1** | 3.1 | 3.2 | **2.1** | 4.8 | 4.7 | **2.2** |
| Compas-R, Race | — | 7.6 | **2.5** | — | 7.9 | **2.6** | — | 9.2 | **2.1** | — | 4.5 | **2.0** |
| Compas-R, Gender | 10.0 | 9.5 | **1.9** | 10.0 | 9.4 | **1.8** | 11.3 | 10.7 | **2.6** | 5.6 | 5.5 | **0.3** |
| Compas-VR, Gender | 14.9 | 12.2 | **2.9** | 8.9 | 10.7 | **2.0** | 14.6 | 10.5 | 7.2 | 12.5 | 10.0 | **1.3** |

Supplement demonstrate the significant gains in accuracy that can be achieved with the proposed approach. We also evaluated the effect of using LLO calibration instead of beta calibration methods and found little difference between the two methods (details in Supplement).

For concreteness we demonstrated our results with three popular fairness metrics ($\Delta$ accuracy, TPR, and FPR) in the paper. However, we can directly extend this approach to handle metrics such as calibration and balance [Kleinberg et al., 2016] as well as ratio-based metrics. In particular, by predicting the distribution of class labels $y$ with the calibrated model scores, any fairness metric that can be defined as a deterministic function of calibrated model scores $s$, labels $y$ and groups $g$ can levarage unlabeled data to reduce variance using our proposed method.

Consideration of the bias-variance properties of the different methods reveals a fundamental tradeoff. The labeled data contribute no bias to the estimate but can have high variance for small $n_L$, whereas the unlabeled data (via their calibrated scores) contribute little variance but can have a persistent bias due to potential misspecification in the parametric calibration model. An open question, that is beyond the scope of this paper, is how to balance this bias-variance tradeoff in a more adaptive fashion as a function of $n_L$ and $n_U$, to further improve the accuracy of estimates of fairness metrics for arbitrary datasets. One potential option would be to a more flexible calibration method (e.g., Gaussian process calibration as proposed in Wenger et al. [2020]). Another option would be to automatically quantify the calibration bias and tradeoff the contributions of labeled and unlabeled data accordingly in estimating $\theta_g$'s and $\Delta$.

We also found empirically that while the posterior credible intervals (CIs) for the BB method are well-calibrated, those for BC tended to be overconfident as $n_L$ increases (see Supplement for details). This is likely due to misspecification in the parametric beta calibration model. An interesting and important direction for future work is to develop methods that are better calibrated in terms of posterior credible intervals (e.g., Syring and Martin [2019]) and that can retain the low-variance advantages of the BC approach we propose here.

# 5 Related Work

Our Bayesian calibration approach builds on the work of Turner et al. [2014] who used hierarchical Bayesian methods for calibration of human judgement data using the LLO calibration model. Bayesian approaches to classifier calibration include marginalizing over binned model scores [Naeini et al., 2015] and calibration based on Gaussian processes [Wenger et al., 2020]. The Bayesian framework of Welinder et al. [2013] in particular is close in spirit to our work in that unlabeled examples are used to improve calibration, but differs in that a generative mixture model is used for modeling of scores rather than direct calibration. None of this prior work on Bayesian calibration addresses fairness assessment and none (apart from Welinder et al. [2013]) leverages unlabeled data.

There has also been work on uncertainty-aware assessment of classifier performance such as the use of Bayesian inference for classifier-related metrics such as marginal accuracy [Benavoli et al., 2017] and precision-recall [Goutte and Gaussier, 2005]. Although these approaches share similarities with our work, they do not make use of unlabeled data. In contrast, the Bayesian evaluation methods proposed by Johnson et al. [2019] can use unlabeled data but makes strong prior assumptions that are specific to the application domain of diagnostic testing. More broadly, other general approaches have been proposed for label-efficient classifier assessment including stratified sampling [Sawade et al., 2010], importance sampling [Kumar and Raj, 2018], and active assessment with Thompson sampling [Ji et al., 2020]. All of these ideas could in principle be used in conjunction with our approach to further reduce estimation error.

In the literature on algorithmic fairness there has been little prior work on uncertainty-aware assessment of fairness metrics—one exception is the proposed use of frequentist confidence interval methods for groupwise fairness in Besse et al. [2018]. Dimitrakakis et al. [2019] proposed a framework called "Bayesian fairness," but focused on decision-theoretic aspects of the problem rather than estimation of metrics. Foulds et al. [2020] developed Bayesian approaches for for parametric smoothing across groups to improve the quality of estimation of intersectional fairness metrics. However, none of this work makes use of unlabeled data to improve fairness assessment. And while there is prior work in fairness on leveraging unlabeled data [Chzhen et al., 2019, Noroozi et al., 2019, Wick et al., 2019, Zhang et al., 2020], the goal of that work has been to produce classifiers that are fair, rather than to assess the fairness of existing classifiers.

Finally, there is recent concurrent work from a frequentist perspective that uses Bernstein inequalities and knowledge of group proportions to upper bound the probability that the difference between the frequentist estimate of $\Delta$ and the true $\Delta$ exceeds some value [Ethayarajh, 2020]. While this work differs from our approach in that it does not explore the use of unlabeled data, the same broad conclusion is reached, namely that there can be high uncertainty in empirical estimates of groupwise fairness metrics, given the typical sizes of datasets used in machine learning.

# 6 Conclusions

To answer to the question "can I trust my fairness metric," we have stressed the importance of being aware of uncertainty in fairness assessment, especially when test sizes are relatively small (as is often the case in practice). To address this issue we propose a framework for combining labeled and unlabeled data to reduce estimation variance, using Bayesian calibration of model scores on unlabeled data. The results demonstrate that the proposed method can systematically produce significantly more accurate estimates of fairness metrics, when compared to only using labeled data, across multiple different classification models, datasets, and sensitive attributes. The framework is straightforward to apply in practice and easy to extend to problems such as intersectional fairness (where estimation uncertainty is likely a significant issue) and to evaluation of fairness-aware algorithms.

## Broader Impacts

Machine learning classifiers are currently widely used to make decisions about individuals, across a broad variety of societal contexts: education admissions, health insurance, medical diagnosis, court decisions, marketing, face recognition, and more—and this trend is likely to continue to grow. It is now well-recognized that these machine learning models are susceptible to built-in biases that can lead to systematic discrimination against protected groups. The machine learning research community has begun to recognize this important issue and in the past few years had devoted considerable research resources towards developing principles, frameworks, and algorithmic solutions to address these problems.

In this general context, this paper addresses the understudied problem of how to assess how fair or unfair a model may be, and how much confidence we should have in this assessment given access to a limited amount of labeled data. One particular example of how the proposed approach can be used is in the increasingly common situation where the user of a blackbox classification model needs to assess its performance from a fairness perspective, in a manner that is separate and independent from the claims made by the entity that trained the model. For example, a hospital system or a university might wish to evaluate the fairness characteristics of a pre-trained classification model in the specific context of the population of patients or students in their institution. The methodology developed in this paper is well-suited to such an application.

In terms of negative potential outcomes, although the proposed approach was shown to be robust across multiple datasets and models relative to existing techniques, as with any machine learning methodology there are nonetheless potential blind spots such as the impact of misspecification in the calibration model on the accuracy of estimates of metrics and on posterior credible intervals. In addition, there is also always the danger of miscommunication of the results of the type of estimation methodology proposed here, in particular given the various challenges in communicating concepts related to uncertainty to a non-expert audience (e.g., van der Bles et al. [2019]).

## Acknowledgments and Disclosure of Funding

This material is based upon work supported in part by the National Science Foundation under grants number 1900644 and 1927245 and by a Qualcomm Faculty Award (PS). In addition this work was partially funded by the Center for Statistics and Applications in Forensic Evidence (CSAFE) through Cooperative Agreement 70NANB20H019 between NIST and Iowa State University, which includes activities carried out at the University of California, Irvine. Additional revenues related to this work include research funding from NIH, NASA, PCORI, SAP and eBay, honoraria for lectures for General Motors, consulting income from Toshiba, and internships at Google and Facebook.

## Footnotes

[1]Stratified sampling by group could help with this issue (e.g., see Sawade et al. [2010]), but stratification might not always be possible in practice, and the total variance will still converge slowly overall.

[2]Note that the term "score" is sometimes defined differently in the calibration literature as the maximum class probability for the model. Both definitions are equivalent mathematically for binary classification.

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
