[Supplementary Material]

# Supplemental Material for "Can I Trust My Fairness Metric? Assessing Fairness with Unlabeled Data and Bayesian Inference"

**Disi Ji**[1]    **Padhraic Smyth**[1]    **Mark Steyvers**[2]
[1]Department of Computer Science    [2]Department of Cognitive Sciences
University of California, Irvine
disij@uci.edu    smyth@ics.uci.edu    mark.steyvers@uci.edu

## Contents

## Appendix: Frequentist Estimation of Group Differences

**Synthetic Example (in Section 1):** In Section 1 in the main paper we described a simple illustrative simulation to emphasize the point that large amounts of labeled data are often necessary to estimate groupwise fairness metrics accurately. The simulation consists of simulated data from two groups, where the underrepresented group makes up 20% of the whole dataset, groupwise positive rates $P(y = 1)$ are both 20%, and the true groupwise TPRs are 95% and 90% (i.e., the true $\Delta$ is 0.05). TPR for group $g$ is defined as $P(\hat{y} = 1 | y = 1, g)$ (See Section 2.1 in the paper for more details on notation). In Figure 1, we show in this simulation that a large number $n_L$ of labeled examples (at least 96,000) is needed to ensure there is a 95% chance that our estimate of the true TPR difference (which is 0.05) lies in the range [0.04, 0.06].

Figure 1: Percentage of 10000 independent simulations whose estimates of $\Delta$ TPR are in the range $[0.04, 0.06]$, as a function of the number of labeled examples $n_L$.

## Appendix: Additional Details on Datasets and Classifiers

**Datasets** We performed experiments with six different real-world datasets. Summary statistics (e.g., test set size) are provided in Table 3 in the main paper. Below we provide additional details about these datasets in terms of background, relevant attributes, and how train and test splits were created. For all datasets, we preprocessed the data using the code from Friedler et al. (2018)[1]. We removed all instances that have missing data, and represented categorical variables with one-hot encoding. As in Friedler (2018), for all datasets except Adult we randomly sampled 2/3 of the data for training and use the remaining 1/3 for test. For the Adult data we re-split the training set of the original data into train and test as in Friedler et al. (2018).

- Adult: The Adult dataset[2] from the UCI Repository of Machine Learning Databases is based on 1994 U.S. census income data. This dataset consists of 14 demographic attributes for individuals. Instances are labeled according to whether their income exceeds $50,000 per year. In our experiments, "race" and "gender" are considered sensitive attributes. Instances are grouped into "Amer-Indian-Inuit," "Asian-Pac-Islander," "Black," "Other" and "White" by race, and "Female" and "Male" by gender. "White" and "Male" are the privileged groups.

- Bank: The Bank dataset[3] contains information about individual collected from a Portuguese banking institution. There are 20 attributes for each individuals, including marital status, education, and type of job. The sensitive attribute we use is "age," binarized by whether a individual's age is above 40 or not. The senior group is considered to be privileged. Instances are labeled by whether the individual has subscribed to a term deposit account or not.

- German: The German Credit dataset[4] from the UCI Repository of Machine Learning Databases describes individuals with 20 attributes including type of housing, credit history status, and employment status. Each instance is labeled as being a good or bad credit risk. The sensitive attributes used are "gender" and "age" (age at least 25 years old) and the privileged groups are defined as "male" and "adult."

- Compas-R: The ProPublica dataset[5] contains information about the use of the COMPAS (Correctional Offender Management Profiling for Alternative Sanctions) risk assessment tool applied to 6,167 individuals in Broward County, Florida. Each individual is labeled by whether they were rearrested within two years after the first arrest. Sensitive attributees are "gender" and "race." By "gender", individuals are grouped into "Male" and "Female"; by "race", individuals are grouped into "Caucasian," "Asian," "Native-American," "African-American," "Hispanic" and "Others." The privileged groups are defined to be "Male" and "Caucasian."

- Compas-VR: This is the violent recidivism version[6] of the ProPublica data (Compas-R above), where the predicted outcome is a re-arrest for a violent crime.

- Ricci: The Ricci dataset[7] is from the case of Ricci v. DeStefano from the Supreme Court of the United States (2009). It contains 118 instances and 5 attributes, including the sensitive attribute "race." The privileged group was defined to be "White." Each instance is labeled by a promotion decision for each individual.

**Classification Models** We used the following classification models in our experiments: logistic regression, multilayer perceptron (MLP) with a single hidden layer of size 10, random forests (the number of trees in the forest is set to 100), Gaussian Naive Bayes. The models were trained using standard default parameter settings and using the code provided by Friedler et al. (2018). Predictions from the trained models were generated on the test data. Sensitive attributes were not included as inputs to the models during training or test.

## Appendix: Complete Experimental Results

In Figure 4 and in Tables 2 and 3 in the main paper we reported summary results of systematic comparisons between the frequentist method, the beta-binomial model (BB) method, and the Bayesian calibration (BC) method, in terms of the mean absolute estimation error as a function of the number of labeled examples $n_L$.

In this section we provide complete tables and graphs for these results. In the tables the lowest error rate per row-column group is in bold if the difference among methods is statistically significant under a Wilcoxon signed-rank test (p=0.05). As in the results in the main paper, the results below demonstrate that BC produces significantly more accurate estimates of group fairness metrics $\Delta$ than the BB or frequentist estimates, across all 4 classification models that we investigated, across FPR, TPR and Accuracy metrics, and across all datasets[8]

Table 1: **MAE for $\Delta$ Accuracy Estimates**, with different $n_L$. Mean absolute error between estimates and true $\Delta$ across 100 runs of labeled samples of different sizes $n_L$ for different trained models (groups of columns) and 10 different dataset-group combinations (groups of rows). Estimation methods are Freq (frequentist), BB (beta-binomial), and BC (Bayesian-calibration). Freq and BB use only labeled samples, BC uses both labeled samples and unlabeled data. Trained models are Multilayer Perceptron, Logistic Regression, Random Forests, and Gaussian NaiveBayes.

| Group | $n$ | Multi-layer Perceptron | | | Logistic Regression | | | Random Forest | | | Gaussian Naive Bayes | | |
|---|---|---|---|---|---|---|---|---|---|---|---|---|---|
| | | Freq | BB | BC | Freq | BB | BC | Freq | BB | BC | Freq | BB | BC |
| Adult | 10 | 16.5 | 18.5 | **3.9** | 16.4 | 18.7 | **2.9** | 16.5 | 18.2 | **3.2** | 17.6 | 18.9 | **3.6** |
| Race | 100 | 8.2 | 8.5 | **3.5** | 7.3 | 9.1 | **3.2** | 7.6 | 9.0 | **3.1** | 8.2 | 9.5 | **2.8** |
| | 1000 | 2.5 | 2.5 | **1.6** | 2.1 | 2.3 | **1.7** | 2.0 | 2.1 | **1.4** | 2.3 | 2.3 | **1.4** |
| Adult | 10 | 19.7 | 17.4 | **5.1** | 19.1 | 16.1 | **2.2** | 17.7 | 17.4 | **4.8** | 19.7 | 16.2 | **5.4** |
| Gender | 100 | 5.5 | 5.4 | 4.4 | 5.6 | 5.5 | **1.9** | 5.9 | 5.9 | **4.1** | 6.2 | 6.0 | **2.7** |
| | 1000 | 1.9 | 1.9 | 1.6 | 1.7 | 1.7 | **1.1** | 1.6 | **1.5** | 2.0 | 1.6 | 1.5 | **1.1** |
| Bank | 10 | 15.9 | 13.9 | **2.5** | 13.9 | 13.0 | **1.4** | 11.8 | 11.1 | **1.0** | 15.5 | 13.7 | **1.7** |
| Age | 100 | 4.4 | 4.3 | **2.0** | 4.3 | 4.3 | **1.2** | 4.3 | 4.2 | **0.9** | 5.0 | 5.0 | **1.1** |
| | 1000 | 1.5 | 1.5 | **1.1** | 1.6 | 1.6 | **0.7** | 1.4 | 1.4 | **0.5** | 1.7 | 1.7 | **0.8** |
| German | 10 | 34.6 | 19.8 | **5.0** | 37.1 | 21.2 | **8.7** | 33.6 | 18.7 | **8.2** | 36.6 | 20.4 | **11.5** |
| age | 100 | 8.5 | 8.0 | **3.9** | 8.2 | 7.6 | **3.8** | 8.8 | 8.2 | **4.3** | 9.7 | 9.1 | **4.2** |
| | 200 | 4.4 | 4.2 | **3.1** | 4.5 | 4.4 | **3.3** | 4.9 | 4.8 | **3.3** | 4.8 | 4.7 | **3.5** |
| German | 10 | 30.7 | 21.6 | **8.2** | 25.6 | 17.4 | **6.3** | 27.7 | 19.3 | **8.6** | 30.0 | 20.1 | **6.5** |
| Gender | 100 | 7.3 | 7.1 | **5.4** | 7.1 | 6.9 | **3.7** | 7.2 | 7.0 | **4.8** | 6.0 | 5.9 | **2.8** |
| | 200 | 3.2 | 3.2 | 3.0 | 4.0 | 3.9 | **2.9** | 3.6 | 3.5 | **2.9** | 4.0 | 4.0 | **2.2** |
| Compas-R | 10 | 31.5 | 21.0 | **4.2** | 31.7 | 20.4 | **4.8** | 29.3 | 20.3 | **2.4** | 33.5 | 23.2 | **8.4** |
| Race | 100 | 6.8 | 6.8 | **2.8** | 7.4 | 7.4 | **3.4** | 8.7 | 8.5 | **1.8** | 8.2 | 7.9 | **6.0** |
| | 1000 | 2.0 | 2.0 | **1.6** | 1.9 | 1.9 | 1.6 | 1.9 | 2.0 | **1.2** | 2.0 | 1.9 | 1.8 |
| Compas-R | 10 | 33.7 | 21.6 | **5.0** | 34.3 | 21.9 | **3.8** | 36.3 | 23.3 | **4.4** | 40.5 | 25.5 | **13.7** |
| Gender | 100 | 9.3 | 8.8 | **3.3** | 9.5 | 9.0 | **2.6** | 8.8 | 8.5 | **2.7** | 10.2 | 9.7 | **8.0** |
| | 1000 | 2.1 | 2.0 | **1.4** | 2.2 | 2.2 | **1.3** | 2.4 | 2.4 | **1.4** | 1.9 | 1.9 | **1.8** |
| Compas-VR | 10 | 18.7 | 17.1 | **4.0** | 18.5 | 15.6 | **4.4** | 18.2 | 15.8 | **2.4** | 26.6 | 19.8 | **6.5** |
| Race | 100 | 5.5 | 5.6 | **3.1** | 5.1 | 5.1 | **3.4** | 6.0 | 6.3 | **2.0** | 6.8 | 6.6 | **3.7** |
| | 1000 | 0.9 | 0.9 | **0.8** | 0.9 | 0.9 | 0.8 | 0.9 | 0.9 | **0.8** | 1.1 | 1.1 | **0.9** |
| Compas-VR | 10 | 20.6 | 16.9 | **5.4** | 19.9 | 16.6 | **5.3** | 22.3 | 19.0 | **6.3** | 31.3 | 21.5 | **9.8** |
| Gender | 100 | 6.4 | 6.3 | **3.4** | 6.1 | 6.0 | **3.1** | 6.3 | 6.3 | **4.4** | 7.3 | 7.1 | **4.5** |
| | 1000 | 1.0 | 1.0 | **0.9** | 1.0 | 1.0 | **0.9** | 0.9 | 0.9 | 1.0 | 1.4 | 1.4 | **0.9** |
| Ricci | 10 | 23.5 | 17.7 | **14.6** | 14.6 | 14.6 | **7.9** | 6.3 | 12.2 | **2.1** | 8.9 | 13.1 | **1.6** |
| Race | 20 | 12.9 | 11.1 | 9.8 | 8.4 | 9.3 | **7.1** | 3.9 | 8.5 | **1.5** | 3.8 | 9.4 | **2.1** |
| | 30 | 8.5 | 7.5 | **6.5** | 4.9 | 5.7 | **4.6** | 2.0 | 6.0 | **1.1** | 2.8 | 6.5 | **2.0** |

Table 2: **MAE for $\Delta$ TPR Estimates**, with different $n_L$. Mean absolute error between estimates and true $\Delta$ across 100 runs of labeled samples of different sizes $n_L$ for different trained models (groups of columns) and 8 different dataset-group combinations (groups of rows). Estimation methods are Freq (Frequentist), BB (beta-binomial), and BC (Bayesian-calibration). Freq and BB use only labeled samples, BC uses both labeled samples and unlabeled data. Trained models are Multilayer Perceptron, Logistic Regression, Random Forests, and Gaussian NaiveBayes.

| Group | $n$ | Multi-layer Perceptron | | | Logistic Regression | | | Random Forest | | | Gaussian Naive Bayes | | |
|---|---|---|---|---|---|---|---|---|---|---|---|---|---|
| | | Freq | BB | BC | Freq | BB | BC | Freq | BB | BC | Freq | BB | BC |
| Adult Race | 40 | — | 16.3 | **7.0** | — | 16.7 | **9.3** | — | 17.9 | **6.2** | — | 23.6 | **4.5** |
| | 100 | — | 15.3 | **6.4** | — | 16.1 | **8.4** | — | 14.9 | **5.6** | — | 20.7 | **3.9** |
| | 200 | — | 12.5 | **5.8** | — | 14.7 | **7.0** | — | 14.3 | **4.6** | — | 14.6 | **3.0** |
| Adult Gender | 40 | — | 21.8 | **5.5** | — | 22.8 | **5.8** | — | 20.9 | **8.4** | — | 21.1 | **11.7** |
| | 100 | — | 17.8 | **5.1** | — | 18.9 | **5.7** | — | 18.6 | **8.4** | — | 17.7 | **11.4** |
| | 200 | 16.3 | 14.3 | **4.3** | 15.8 | 14.0 | **4.6** | 16.1 | 14.2 | **7.3** | 15.0 | 13.4 | 11.5 |
| Bank Age | 40 | — | 24.2 | **6.1** | — | 25.4 | **3.8** | — | 25.2 | **2.7** | — | 23.0 | **3.6** |
| | 100 | 25.9 | 20.0 | **5.0** | 25.7 | 20.4 | **4.0** | 20.9 | 16.6 | **2.8** | 24.9 | 19.6 | **2.6** |
| | 200 | 16.8 | 15.0 | **4.8** | 17.7 | 15.9 | **4.2** | 16.6 | 14.9 | **3.1** | 17.3 | 15.7 | **2.3** |
| German age | 40 | — | 15.0 | **3.9** | — | 18.4 | **3.0** | — | 11.3 | **3.6** | — | 16.7 | **6.3** |
| | 100 | 8.9 | 8.0 | **3.5** | 10.7 | 9.7 | **3.1** | 8.0 | 7.1 | **3.5** | 12.9 | 11.5 | **3.3** |
| | 200 | 4.7 | 4.7 | **3.0** | 5.6 | 5.4 | **2.6** | 5.1 | 5.1 | **3.1** | 6.8 | 6.5 | **2.8** |
| German Gender | 40 | 2.6 | 4.5 | **2.3** | 11.8 | 10.0 | **2.4** | 9.4 | 8.1 | **2.4** | 15.0 | 13.1 | **3.8** |
| | 100 | 1.4 | 2.1 | 2.0 | 6.5 | 6.3 | **2.1** | 5.9 | 5.8 | **2.3** | 7.7 | 7.4 | **3.1** |
| | 200 | **0.7** | 1.0 | 1.6 | 3.3 | 3.3 | **2.1** | 3.1 | 3.2 | **2.1** | 4.8 | 4.7 | **2.2** |
| Compas-R Race | 40 | — | 15.2 | **3.4** | — | 16.3 | **3.4** | — | 14.8 | **3.2** | — | 10.1 | **2.2** |
| | 100 | — | 11.5 | **2.9** | — | 11.5 | **3.1** | — | 10.6 | **2.5** | — | 6.7 | **2.1** |
| | 200 | — | 7.6 | **2.5** | — | 7.9 | **2.6** | — | 9.2 | **2.1** | — | 4.5 | **2.0** |
| Compas-R Gender | 40 | — | 19.3 | **2.7** | — | 21.8 | **2.5** | — | 19.3 | **3.4** | — | 14.0 | **0.1** |
| | 100 | 15.9 | 13.7 | **2.4** | 17.6 | 15.1 | **2.1** | 14.3 | 12.5 | **3.2** | 8.7 | 8.0 | **0.2** |
| | 200 | 10.0 | 9.5 | **1.9** | 10.0 | 9.4 | **1.8** | 11.3 | 10.7 | **2.6** | 5.6 | 5.5 | **0.3** |
| Compas-VR Gender | 40 | — | 23.0 | **3.8** | — | 27.0 | **2.2** | — | 20.9 | **9.0** | — | 21.1 | **1.2** |
| | 100 | — | 18.0 | **3.2** | — | 19.7 | **2.1** | — | 16.3 | **8.1** | — | 14.9 | **1.2** |
| | 200 | 14.9 | 12.2 | **2.9** | 8.9 | 10.7 | **2.0** | 14.6 | 10.5 | 7.2 | 12.5 | 10.0 | **1.3** |

Table 3: **MAE for $\Delta$ FPR Estimates**, with different $n_L$. Same setup as Table 2.

| Group | $n$ | Multi-layer Perceptron | | | Logistic Regression | | | Random Forest | | | Gaussian Naive Bayes | | |
|---|---|---|---|---|---|---|---|---|---|---|---|---|---|
| | | Freq | BB | BC | Freq | BB | BC | Freq | BB | BC | Freq | BB | BC |
| Adult Race | 40 | — | 16.1 | **1.5** | — | 16.6 | **1.5** | — | 16.7 | **1.8** | — | 16.5 | **2.9** |
| | 100 | — | 9.6 | **1.2** | — | 10.1 | **1.5** | — | 10.5 | **1.7** | — | 10.7 | **2.8** |
| | 200 | — | 5.7 | **1.2** | — | 6.3 | **1.6** | — | 6.4 | **1.8** | — | 6.6 | 3.4 |
| Adult Gender | 40 | 7.1 | 6.9 | **2.6** | 7.2 | 7.1 | **2.6** | 8.3 | 8.1 | **3.7** | 10.3 | 9.8 | **5.1** |
| | 100 | 4.4 | 4.3 | **2.3** | 4.3 | 4.1 | **2.2** | 5.2 | 5.1 | **3.5** | 6.6 | 6.4 | **4.7** |
| | 200 | 3.2 | 3.3 | **2.5** | 3.2 | 3.2 | **2.3** | 3.7 | 3.7 | **3.4** | 4.7 | 4.6 | **4.6** |
| Bank Age | 40 | 2.4 | 2.5 | **0.5** | 3.6 | 3.7 | **0.6** | 4.1 | 4.2 | **0.7** | 8.5 | 7.9 | **1.3** |
| | 100 | 1.9 | 1.8 | **0.5** | 2.4 | 2.4 | **0.6** | 3.3 | 3.3 | **0.7** | 5.3 | 5.2 | **1.3** |
| | 200 | 1.5 | 1.5 | **0.5** | 2.1 | 2.0 | **0.6** | 2.1 | 2.1 | **0.7** | 3.6 | 3.6 | **1.3** |
| German age | 40 | — | 19.7 | **9.8** | — | 18.4 | **8.7** | — | 18.7 | **9.1** | — | 17.6 | **13.2** |
| | 100 | 16.6 | 14.3 | **7.4** | 13.6 | 11.9 | **6.3** | 13.7 | 11.7 | **6.3** | 14.9 | 12.5 | **12.0** |
| | 200 | 8.6 | 8.0 | **5.6** | 7.7 | 7.2 | **5.7** | 7.2 | 6.8 | **5.4** | 8.4 | **7.7** | 8.3 |
| German Gender | 40 | 15.6 | 13.2 | **6.8** | 27.3 | 21.5 | **5.3** | 23.1 | 18.6 | **8.4** | 20.3 | 16.1 | **4.4** |
| | 100 | 9.2 | 9.0 | **5.7** | 14.4 | 13.2 | **5.9** | 13.3 | 12.4 | **7.4** | 12.6 | 11.7 | **5.6** |
| | 200 | 4.9 | 4.9 | **3.8** | 7.3 | 7.0 | **5.2** | 6.8 | 6.6 | **5.0** | 5.9 | 5.7 | **4.6** |
| Compas-R Race | 40 | — | 15.1 | **3.7** | — | 13.2 | **3.3** | — | 14.5 | **5.6** | — | 10.8 | **6.2** |
| | 100 | — | 8.4 | **2.7** | — | 8.5 | **2.4** | — | 10.0 | **4.6** | — | 7.3 | **4.4** |
| | 200 | — | 6.8 | **2.1** | — | 5.9 | **1.9** | — | 6.7 | **3.7** | — | 4.9 | **3.4** |
| Compas-R Gender | 40 | — | 13.5 | **3.4** | — | 15.0 | **2.4** | — | 16.2 | **4.9** | — | 10.9 | **4.2** |
| | 100 | 7.7 | 7.4 | **3.2** | 8.5 | 8.3 | **2.4** | 11.5 | 11.0 | **5.0** | 7.4 | 6.9 | **5.3** |
| | 200 | 5.3 | 5.2 | **2.7** | 6.1 | 6.1 | **2.0** | 8.5 | 8.4 | **4.4** | 5.1 | **5.0** | 5.6 |
| Compas-VR Gender | 40 | 5.6 | 6.6 | **0.7** | 3.3 | 5.6 | **0.4** | 5.5 | 7.5 | **1.9** | 12.8 | 11.7 | **4.4** |
| | 100 | 4.0 | 4.3 | **0.6** | 2.4 | 2.8 | **0.4** | 3.9 | 4.4 | **1.5** | 6.3 | 6.3 | **4.8** |
| | 200 | 2.5 | 2.6 | **0.5** | 1.8 | 1.8 | **0.4** | 2.5 | 2.6 | **1.2** | 5.1 | 4.9 | **4.1** |

Figure 2: **MAE for Accuracy:** Mean absolute error (MAE) of the difference between algorithm estimates and ground truth for group difference in accuracy across 100 runs, as a function of number of labeled instances, for 10 different dataset-group pairs and 4 classifiers. Shading indicates 95% error bars for each method (not shown for the frequentist curve to avoid overplotting). Upper right corner shows the ground truth Δ between the unprivileged group and the privileged group.

Figure 3: **MAE for TPR:** Mean absolute error (MAE) of the difference between algorithm estimates and ground truth for group difference in TPR across 100 runs. Compas-VR race and Ricci race are not included since there are no positive instances for some groups. Same setup as Figure 2.

Figure 4: **MAE for FPR:** Mean absolute error (MAE) of the difference between algorithm estimates and ground truth for groupwise difference in FPR across 100 runs. Same setup as Figure 3.

## Appendix: Calibration Coverage of Posterior Credible Intervals

We can generate posterior credible intervals on $\Delta$ (as shown in red in Figure 3 in the main paper) for both the BB and BC methods by computing upper and lower percentiles from posterior samples for $\Delta$. Below in Table 4 we show the coverage of 95% credible intervals for both the BB (beta-bernoulli) and BC (Bayesian-calibration) methods, for the multi-layer perceptron model. Coverage is defined as the percentage of credible intervals (across multiple different labeled datasets of size $n_L$) that contain the true value: a perfectly calibrated 95% credible interval would have 95% coverage. Table 4 shows that while the coverage for both methods is generally not far from 95% there is room for improvement (as discussed in the main paper). For example, for small values of $n_L$ the coverage of both methods is often too high (above 95%), with some evidence of coverage decreasing as $n_L$ increasing. Generating accurate posterior credible intervals is a known issue in Bayesian analysis in the presence of model misspecification (e.g., Syring and Martin (2019)) and is an interesting direction for future work on Bayesian analysis of fairness metrics.

Table 4: **Calibration Coverage of Posterior Credible Intervals Comparison**, across 1000 runs of labeled samples of different sizes $n_L$ for 10 different dataset-group combinations (rows). Estimation methods are BC (Bayesian-Calibration) and BB (beta-bernoulli). Trained model is Multilayer Perceptron.

| Group | $n_L = 10$ BC | $n_L = 10$ BB | $n_L = 20$ BC | $n_L = 20$ BB | $n_L = 40$ BC | $n_L = 40$ BB | $n_L = 100$ BC | $n_L = 100$ BB |
|---|---|---|---|---|---|---|---|---|
| Adult, Race | 99.9 | 97.7 | 98.6 | 93.5 | 96.2 | 93.2 | 92.3 | 95.3 |
| Adult, Gender | 100.0 | 96.4 | 99.7 | 95.5 | 99.2 | 94.9 | 96.8 | 95.5 |
| Bank, Age | 99.4 | 98.7 | 98.8 | 98.5 | 98.0 | 96.4 | 93.7 | 95.3 |
| German, age | 99.9 | 98.8 | 99.6 | 98.1 | 99.0 | 98.3 | 96.9 | 98.3 |
| German, Gender | 99.1 | 97.4 | 99.1 | 97.4 | 97.7 | 96.4 | 94.6 | 97.8 |
| Compas-R, Race | 99.3 | 98.8 | 99.4 | 97.2 | 99.1 | 96.7 | 99.3 | 96.6 |
| Compas-R, Gender | 99.3 | 97.7 | 99.3 | 97.0 | 98.6 | 95.9 | 97.6 | 96.5 |
| Compas-VR, Race | 99.6 | 100.0 | 98.6 | 97.8 | 97.9 | 95.2 | 97.5 | 93.1 |
| Compas-VR, Gender | 96.3 | 97.2 | 94.3 | 96.5 | 95.4 | 96.1 | 95.8 | 97.1 |
| Ricci, Race | 93.2 | 99.7 | 91.4 | 99.7 | — | — | — | — |

# Appendix: Graphical Model for Hierarchical Beta Calibration

Figure 5: Graphical model for hierarchical beta calibration as described in Section 2.4 of the main paper. $\Gamma$ is the hyperprior on $\pi$, representing the fixed parameters for the normal and truncated normal hyperpriors described in Section 2.4 in the main paper.

## Appendix: Error Results with Non-Hierarchical Bayesian Calibration

In our Hierarchical Bayesian calibration model we allows different groups to share statistical strength via a hierarchical structure. In this section, we compare our proposed Bayesian calibration model (BC) that uses this hierarchy with a non-hierarchical Bayesian calibration (NHBC) approach. Table 5 compares the mean absolute error (MAE) rate for both approaches in estimating differences in accuracy between groups (same setup as Tables 2 and 3 in the main paper in terms of how MAE is computed). The results show that (1) both BC and NHBC significantly improve MAE compared to BB; (2) BC and NHBC are comparable in most cases, but with the hierarchical structure the BC method avoids occasional catastrophic errors that NHBC can make, e.g. when assessing $\Delta$ Accuracy of a Gaussian Naive Bayes model on Compas-R Gender and Compas-VR Gender.

Table 5: **MAE for $\Delta$ Accuracy Estimates**, with different $n_L$. Mean absolute error between estimates and true $\Delta$ across 100 runs of labeled samples of different sizes $n_L$ for different trained models (groups of columns) and 10 different dataset-group combinations (groups of rows). Estimation methods are BB (beta-binomial), and NHBC (non-hierarchical Bayesian-calibration), BC (Bayesian-Calibration). BB uses only labeled samples, NHBC and BC use both labeled samples and unlabeled data. Trained models are Multilayer Perceptron, Logistic Regression, Random Forests, and Gaussian NaiveBayes.

| Group | $n$ | Multi-layer Perceptron | | | Logistic Regression | | | Random Forest | | | Gaussian Naive Bayes | | |
|---|---|---|---|---|---|---|---|---|---|---|---|---|---|
| | | BB | NHBC | BC | BB | NHBC | BC | BB | NHBC | BC | BB | NHBC | BC |
| Adult Race | 10 | 18.4 | **3.2** | 3.9 | 18.8 | **2.7** | 2.9 | 18.1 | **2.8** | 3.2 | 18.9 | 4.5 | **3.6** |
| | 20 | 16.1 | **3.3** | 4.4 | 16.7 | **2.9** | 3.4 | 16.3 | **3.0** | 3.7 | 16.8 | 4.1 | **3.7** |
| | 40 | 13.1 | **2.8** | 4.5 | 14.0 | **2.9** | 3.7 | 14.4 | **2.9** | 3.8 | 14.4 | 3.7 | **3.3** |
| | 100 | 8.6 | **2.7** | 3.5 | 9.2 | **3.0** | 3.2 | 9.0 | **2.6** | 3.1 | 9.6 | **2.4** | 2.8 |
| | 1000 | 2.5 | **1.4** | 1.6 | 2.3 | 2.1 | **1.7** | 2.1 | **0.7** | 1.4 | 2.3 | 1.8 | **1.4** |
| Adult Gender | 10 | 17.4 | **4.1** | 5.1 | 16.3 | 2.6 | **2.2** | 17.3 | 5.3 | **4.8** | 16.3 | 7.2 | **5.4** |
| | 20 | 12.9 | **4.4** | 5.1 | 12.2 | 2.6 | **2.2** | 12.4 | 5.3 | **4.9** | 11.6 | 6.7 | **4.5** |
| | 40 | 9.0 | **4.1** | 4.9 | 9.2 | 2.5 | **2.1** | 9.6 | 5.1 | **4.5** | 9.7 | 6.3 | **3.9** |
| | 100 | 5.4 | **3.1** | 4.4 | 5.5 | 2.0 | **2.0** | 5.9 | 4.7 | **4.1** | 6.0 | 4.8 | **2.7** |
| | 1000 | 1.9 | **1.4** | 1.6 | 1.7 | **1.0** | 1.1 | **1.5** | 1.8 | 2.0 | 1.5 | **0.9** | 1.0 |
| Bank Age | 10 | 14.0 | **1.7** | 2.5 | 12.8 | 1.5 | **1.4** | 11.2 | 1.1 | **1.0** | 13.7 | **1.4** | 1.7 |
| | 20 | 11.6 | **2.3** | 2.9 | 10.9 | 1.9 | **1.7** | 8.8 | 1.4 | **1.2** | 10.3 | **1.6** | 1.7 |
| | 40 | 8.0 | **2.3** | 2.6 | 7.3 | 1.7 | **1.4** | 6.5 | 1.5 | **1.1** | 7.5 | 1.7 | **1.5** |
| | 100 | 4.3 | 2.2 | **2.0** | 4.3 | 1.4 | **1.2** | 4.2 | 1.2 | **0.9** | 4.9 | 1.3 | **1.1** |
| | 1000 | 1.5 | 1.2 | **1.1** | 1.6 | 0.8 | **0.7** | 1.4 | 0.6 | **0.5** | 1.7 | **0.7** | 0.8 |
| German age | 10 | 19.7 | 5.6 | **5.0** | 21.3 | 10.3 | **8.7** | 19.1 | **8.2** | 8.2 | 20.4 | 14.2 | **11.5** |
| | 20 | 18.1 | 6.0 | **4.4** | 18.6 | 6.7 | **6.4** | 16.7 | **7.0** | 7.0 | 18.8 | 9.9 | **9.0** |
| | 40 | 15.9 | 6.7 | **4.8** | 15.0 | 5.6 | **4.9** | 11.7 | 6.6 | **5.8** | 14.9 | **6.4** | 6.9 |
| | 100 | 7.9 | 5.8 | **3.9** | 7.5 | 5.5 | **3.8** | 8.2 | 6.5 | **4.3** | 9.1 | 4.4 | **4.2** |
| | 200 | 4.2 | 3.7 | **3.1** | 4.4 | 4.1 | **3.3** | 4.7 | 4.1 | **3.3** | 4.7 | 3.8 | **3.5** |
| German Gender | 10 | 21.5 | 10.5 | **8.2** | 17.6 | 7.0 | **6.3** | 19.4 | **8.5** | 8.6 | 20.0 | **5.9** | 6.5 |
| | 20 | 16.2 | 10.0 | **7.8** | 13.2 | 7.1 | **5.1** | 14.1 | 8.4 | **7.8** | 15.4 | 5.9 | **4.9** |
| | 40 | 11.6 | 9.2 | **6.6** | 11.4 | 8.4 | **4.5** | 11.1 | 7.7 | **5.9** | 11.1 | 6.1 | **3.8** |
| | 100 | 7.1 | 6.5 | **5.4** | 6.9 | 6.6 | **3.7** | 7.0 | 6.1 | **4.8** | 5.9 | 6.4 | **2.8** |
| | 200 | 3.2 | 3.3 | **3.0** | 4.0 | 4.0 | **2.9** | 3.6 | 3.4 | **2.9** | 4.0 | 4.0 | **2.2** |
| Compas-R Race | 10 | 21.1 | **2.9** | 4.2 | 20.7 | **4.0** | 4.8 | 20.3 | **1.4** | 2.4 | 23.1 | **6.6** | 8.4 |
| | 20 | 14.8 | **2.8** | 3.3 | 15.2 | 3.9 | **3.8** | 15.8 | **2.0** | 2.5 | 16.6 | **7.8** | 8.0 |
| | 40 | 11.7 | 3.0 | **3.0** | 12.1 | 3.9 | **3.6** | 11.6 | 2.0 | **2.0** | 10.9 | 9.9 | **8.1** |
| | 100 | 6.8 | 2.9 | **2.8** | 7.4 | 3.7 | **3.4** | 8.5 | 2.1 | **1.8** | 7.9 | 7.7 | **6.0** |
| | 1000 | 2.0 | **1.5** | 1.6 | 1.9 | **1.6** | 1.7 | 1.9 | 1.3 | **1.2** | 1.9 | 1.9 | **1.8** |
| Compas-R Gender | 10 | 21.3 | **3.8** | 5.0 | 22.0 | **3.4** | 3.8 | 23.4 | **3.5** | 4.4 | 25.4 | 19.1 | **13.7** |
| | 20 | 18.5 | **3.8** | 5.1 | 18.4 | **3.3** | 4.0 | 17.4 | **3.3** | 4.6 | 21.4 | 23.8 | **12.3** |
| | 40 | 12.2 | **3.4** | 4.0 | 13.0 | **3.0** | 3.3 | 13.7 | **2.8** | 3.6 | 15.0 | 23.8 | **9.5** |
| | 100 | 8.8 | **3.2** | 3.3 | 9.1 | 2.7 | **2.6** | 8.5 | **2.1** | 2.7 | 9.8 | 15.5 | **8.0** |
| | 1000 | 2.0 | 1.7 | **1.4** | 2.2 | 1.4 | **1.3** | 2.4 | 1.6 | **1.4** | 1.9 | 1.9 | **1.8** |
| Compas-VR Race | 10 | 17.4 | 4.0 | **4.0** | 15.6 | **4.4** | 4.4 | 15.7 | 2.6 | **2.4** | 19.7 | **6.1** | 6.5 |
| | 20 | 13.5 | 4.7 | **4.3** | 13.7 | 5.0 | **4.8** | 13.6 | 3.3 | **2.9** | 15.9 | 10.7 | **6.5** |
| | 40 | 9.6 | 4.5 | **3.8** | 9.6 | 4.5 | **3.9** | 9.9 | 3.1 | **2.4** | 11.1 | 8.8 | **5.5** |
| | 100 | 5.6 | 3.6 | **3.1** | 5.2 | 3.8 | **3.4** | 6.2 | 2.6 | **2.0** | 6.6 | 6.8 | **3.7** |
| | 1000 | 0.9 | 0.8 | **0.8** | 0.9 | **0.8** | 0.8 | 0.9 | 0.8 | **0.8** | 1.1 | 1.2 | **0.9** |
| Compas-VR Gender | 10 | 17.2 | 5.6 | **5.4** | 16.8 | 5.7 | **5.3** | 19.0 | **5.8** | 6.3 | 21.3 | 18.9 | **9.8** |
| | 20 | 13.3 | 5.4 | **5.1** | 14.1 | 5.4 | **4.9** | 14.0 | **5.7** | 6.2 | 16.0 | 28.2 | **8.7** |
| | 40 | 9.3 | 5.1 | **4.7** | 9.7 | 4.9 | **4.5** | 10.5 | **5.3** | 5.7 | 12.4 | 30.9 | **6.9** |
| | 100 | 6.4 | 3.7 | **3.4** | 5.9 | 3.5 | **3.1** | 6.3 | **4.2** | 4.4 | 7.1 | 18.5 | **4.5** |
| | 1000 | 1.0 | **0.8** | 0.9 | 1.0 | 0.9 | **0.9** | 0.9 | **0.9** | 1.0 | 1.4 | **0.9** | 0.9 |
| Ricci Race | 10 | 17.7 | 16.1 | **14.6** | 14.4 | **7.5** | 7.9 | 12.2 | **1.9** | 2.1 | 13.1 | 1.7 | **1.6** |
| | 20 | 11.2 | 11.8 | **9.8** | 9.3 | 7.2 | **7.1** | 8.5 | **1.5** | 1.5 | 9.5 | **2.0** | 2.1 |
| | 30 | 7.4 | 7.7 | **6.5** | 5.8 | 5.1 | **4.6** | 6.0 | 1.1 | **1.1** | 6.4 | **1.9** | 2.0 |

## Appendix: Sensitivity Analysis for Calibration Priors

As described in Section 2.4, we use the beta calibration model to recalibrate a model score $s$ for the $g$-th group according to

$$f(s; a_g, b_g, c_g) = \frac{1}{1 + e^{-c_g - a_g \log s + b_g \log(1-s)}}$$

where $a_g$, $b_g$, and $c_g$ are calibration parameters with $a_g, b_g \geq 0$. With $a_g = 1, b_g = 1, c_g = 0$, $f(\cdot; 1, 1, 0)$ is an identity function. We assume that the parameters from each individual group are sampled from a shared distribution:

$$\log a_g \sim \mathrm{N}(\mu_a, \sigma_a), \log b_g \sim \mathrm{N}(\mu_b, \sigma_b), c_g \sim \mathrm{N}(\mu_c, \sigma_c)$$

where $\pi = \{\mu_a, \sigma_a, \mu_b, \sigma_b, \mu_c, \sigma_c\}$ is the set of hyperparameters of the shared distributions. As discussed in the main paper, in our experiments we set the hyperparameters as

$$\mu_a \sim \mathrm{N}(0, .4), \mu_b \sim \mathrm{N}(0, .4), \mu_c \sim \mathrm{N}(0, 2), \sigma_a \sim \mathrm{TN}(0, .15), \sigma_b \sim \mathrm{TN}(0, .15), \sigma_c \sim \mathrm{TN}(0, .75)$$

These prior distributions encode a weak prior belief that the model scores are calibrated by placing the mode of $a_g, b_g$ and $c_g$ at 1, 1, and 0 respectively. We used exactly these prior settings in all our experiments across all datasets, all groups, and all labeled and unlabeled dataset sizes, which already demonstrates to a certain extent the robustness of these settings.

In this Appendix we describe the results of a sensitivity analysis with respect to the variances in the prior above. We evaluate our proposed methodology over a range of settings for the variances, multiplying the default values with different values of $\alpha$, i.e.

$$\mu_a \sim \mathrm{N}(0, .4\alpha), \sigma_a \sim \mathrm{TN}(0, .15\alpha)$$
$$\mu_b \sim \mathrm{N}(0, .4\alpha), \sigma_b \sim \mathrm{TN}(0, .15\alpha)$$
$$\mu_c \sim \mathrm{N}(0, 2\alpha), \sigma_c \sim \mathrm{TN}(0, .75\alpha)$$

with $\alpha$ ranging from 0.1 to 10. We reran our analysis, using the different variance settings, for the specific case of estimating the change $\Delta$ in accuracy estimates for the Adult dataset grouped by the attribute "race," for each of the four classification models in our study and with different amounts of labeled data.

Table 6 shows the resulting MAE values as $\alpha$ is varied. The results show that the Bayesian calibration (BC) model is robust to the settings of prior variances. Specifically, as $\alpha$ varies from 0.1 to 10 the MAE values with BC are almost always smaller than the ones obtained with BB, and there is a broad range of values $\alpha$ where the MAE values are close to their minimum The results also show that the BC method has less sensitivity to $\alpha$ when the number of labeled examples $n_L$ is large, e.g. $n_L = 1000$.

Table 6: **MAE for $\Delta$ Accuracy Estimates** of the adult data grouped by attribute "race," with different values of $n_L$. Shown are mean absolute error (MAE) values between estimates and true $\Delta$ across 100 runs of labeled samples of different sizes $n_L$ for different trained models (groups of columns). Estimation methods are BB (beta-binomial) and BC (Bayesian-calibration) with different values of $\alpha$ (rows). BB uses only labeled samples, and BC use both labeled samples and unlabeled data.

| Method | Multi-layer Perceptron | | | Logistic Regression | | | Random Forest | | | Gaussian Naive Bayes | | |
|---|---|---|---|---|---|---|---|---|---|---|---|---|
| | 10 | 100 | 1000 | 10 | 100 | 1000 | 10 | 100 | 1000 | 10 | 100 | 1000 |
| BB | 18.52 | 8.48 | 2.46 | 18.74 | 9.14 | 2.30 | 18.24 | 9.00 | 2.12 | 18.88 | 9.54 | 2.32 |
| BC, $\alpha$=0.1 | 2.63 | 2.60 | 2.27 | 2.46 | 2.49 | 2.13 | 2.87 | 2.84 | 2.43 | 4.67 | 4.51 | 0.78 |
| BC, $\alpha$=0.2 | 2.63 | 2.56 | 2.08 | 2.46 | 2.51 | 2.06 | 2.85 | 2.83 | 2.09 | 4.63 | 3.95 | 0.82 |
| BC, $\alpha$=0.3 | 2.60 | 2.52 | 1.88 | 2.42 | 2.51 | 1.95 | 2.85 | 2.79 | 1.86 | 4.44 | 3.36 | 0.97 |
| BC, $\alpha$=0.4 | 2.49 | 2.46 | 1.74 | 2.41 | 2.57 | 1.90 | 2.74 | 2.82 | 1.70 | 4.25 | 3.06 | 1.11 |
| BC, $\alpha$=0.5 | 2.49 | 2.38 | 1.71 | 2.44 | 2.60 | 1.82 | 2.82 | 2.77 | 1.65 | 4.01 | 2.86 | 1.43 |
| BC, $\alpha$=0.6 | 2.47 | 2.37 | 1.62 | 2.55 | 2.62 | 1.75 | 2.82 | 2.88 | 1.60 | 3.81 | 2.79 | 1.46 |
| BC, $\alpha$=0.7 | 2.61 | 2.48 | 1.51 | 2.36 | 2.63 | 1.70 | 2.90 | 2.86 | 1.54 | 3.54 | 2.80 | 1.50 |
| BC, $\alpha$=0.8 | 2.86 | 2.30 | 1.47 | 2.52 | 2.73 | 1.63 | 2.87 | 2.86 | 1.46 | 3.51 | 2.77 | 1.60 |
| BC, $\alpha$=0.9 | 2.93 | 2.27 | 1.43 | 2.44 | 2.82 | 1.64 | 2.87 | 2.90 | 1.46 | 3.14 | 2.91 | 1.58 |
| BC, $\alpha$=1.0 | 3.05 | 2.31 | 1.50 | 2.71 | 2.74 | 1.57 | 2.99 | 2.96 | 1.42 | 3.31 | 2.85 | 1.68 |
| BC, $\alpha$=1.1 | 3.14 | 2.37 | 1.45 | 2.65 | 2.86 | 1.55 | 2.90 | 3.10 | 1.40 | 3.25 | 3.03 | 1.65 |
| BC, $\alpha$=1.2 | 3.11 | 2.19 | 1.49 | 2.73 | 2.80 | 1.52 | 3.27 | 3.01 | 1.39 | 3.20 | 3.03 | 1.68 |
| BC, $\alpha$=1.3 | 3.48 | 2.30 | 1.51 | 2.91 | 2.94 | 1.54 | 3.11 | 3.21 | 1.39 | 3.15 | 2.96 | 1.71 |
| BC, $\alpha$=1.4 | 3.76 | 2.28 | 1.47 | 3.17 | 3.01 | 1.51 | 3.26 | 3.21 | 1.30 | 3.48 | 3.21 | 1.75 |
| BC, $\alpha$=1.5 | 3.67 | 2.20 | 1.49 | 3.12 | 2.94 | 1.51 | 3.46 | 3.05 | 1.34 | 3.23 | 3.19 | 1.66 |
| BC, $\alpha$=1.6 | 4.06 | 2.24 | 1.45 | 3.26 | 2.93 | 1.47 | 3.56 | 3.13 | 1.33 | 3.48 | 3.17 | 1.69 |
| BC, $\alpha$=1.7 | 4.02 | 2.27 | 1.46 | 3.46 | 3.15 | 1.46 | 3.75 | 3.10 | 1.27 | 3.43 | 3.19 | 1.74 |
| BC, $\alpha$=1.8 | 4.35 | 2.14 | 1.42 | 3.36 | 3.09 | 1.50 | 3.76 | 3.26 | 1.29 | 3.67 | 3.22 | 1.81 |
| BC, $\alpha$=1.9 | 4.35 | 2.30 | 1.48 | 3.48 | 2.94 | 1.42 | 3.54 | 3.30 | 1.28 | 3.82 | 3.35 | 1.84 |
| BC, $\alpha$=2.0 | 4.69 | 2.16 | 1.44 | 3.87 | 2.99 | 1.54 | 3.91 | 3.46 | 1.21 | 3.83 | 3.18 | 1.81 |
| BC, $\alpha$=5.0 | 8.11 | 2.54 | 1.63 | 6.31 | 3.32 | 1.53 | 5.32 | 4.13 | 1.31 | 5.25 | 3.82 | 2.13 |
| BC, $\alpha$=10.0 | 10.39 | 2.63 | 1.63 | 7.18 | 3.83 | 1.70 | 7.19 | 4.41 | 1.42 | 6.32 | 4.08 | 2.33 |

# Appendix: Error Results with LLO Calibration

Our hierarchical Bayesian calibration approach can be adapted to use other parametric calibration methods. In addition to the beta calibration method described in the main paper, we also experimented with LLO (linear in log odds) calibration.

Table 7 below shows a direct comparison of the mean absolute error (MAE) rate for estimation of differences in accuracy between groups (same setup as Tables 2 and 3 in the main paper in terms of how MAE is computed). The results show that in general the MAE of the two calibration methods tends to be very similar (relative to the size of the BB and frequentist MAEs) across different dataset-attribute combinations. different prediction models, and different $n_L$ values.

Table 7: **MAE for $\Delta$ Accuracy Estimates of LLO and BC**, with different $n_L$. Mean absolute error between estimates and true $\Delta$ across 100 runs of labeled samples of different sizes $n_L$ for different trained models (groups of columns) and 10 different dataset-group combinations (groups of rows). Estimation methods are BC (Bayesian-Calibration) and LLO (Linear in Log Odds Calibration). Both methods use both labeled samples and unlabeled data. Trained models are Multilayer Perceptron, Logistic Regression, Random Forests, and Gaussian NaiveBayes.

| Group | $n$ | Multi-layer Perceptron | | Logistic Regression | | Random Forest | | Gaussian Naive Bayes | |
|---|---|---|---|---|---|---|---|---|---|
| | | BC | LLO | BC | LLO | BC | LLO | BC | LLO |
| Adult | 10 | 3.9 | 3.8 | 2.9 | 2.8 | 3.2 | 3.2 | 3.6 | 3.5 |
| Race | 100 | 3.5 | 3.4 | 3.2 | 3.1 | 3.1 | 2.9 | 2.8 | 2.4 |
| | 1000 | 1.6 | 2.3 | 1.7 | 2.0 | 1.4 | 1.5 | 1.4 | 1.6 |
| Adult | 10 | 5.1 | 5.1 | 2.2 | 2.3 | 4.8 | 4.7 | 5.4 | 5.0 |
| Gender | 100 | 4.4 | 4.3 | 1.9 | 2.0 | 4.1 | 3.7 | 2.7 | 2.7 |
| | 1000 | 1.6 | 2.2 | 1.1 | 1.0 | 2.0 | 1.5 | 1.1 | 1.1 |
| Bank | 10 | 2.5 | 2.3 | 1.4 | 1.2 | 1.0 | 0.9 | 1.7 | 1.7 |
| Age | 100 | 2.0 | 2.0 | 1.2 | 1.2 | 0.9 | 0.9 | 1.1 | 1.2 |
| | 1000 | 1.1 | 1.2 | 0.7 | 0.7 | 0.5 | 0.5 | 0.8 | 0.9 |
| German | 10 | 5.0 | 4.6 | 8.7 | 8.0 | 8.2 | 7.5 | 11.5 | 10.7 |
| age | 100 | 3.9 | 4.1 | 3.8 | 4.7 | 4.3 | 4.0 | 4.2 | 6.0 |
| | 200 | 3.1 | 3.9 | 3.3 | 4.2 | 3.3 | 3.1 | 3.5 | 6.0 |
| German | 10 | 8.2 | 6.4 | 6.3 | 5.0 | 8.6 | 6.9 | 6.5 | 5.3 |
| Gender | 100 | 5.4 | 5.1 | 3.7 | 3.6 | 4.8 | 4.5 | 2.8 | 3.1 |
| | 200 | 3.0 | 3.4 | 2.9 | 2.8 | 2.9 | 3.1 | 2.2 | 2.9 |
| Compas-R | 10 | 4.2 | 4.6 | 4.8 | 5.2 | 2.4 | 2.5 | 8.4 | 8.2 |
| Race | 100 | 2.8 | 4.4 | 3.4 | 4.8 | 1.8 | 1.4 | 6.0 | 5.6 |
| | 1000 | 1.6 | 5.0 | 1.6 | 4.4 | 1.2 | 1.1 | 1.8 | 2.9 |
| Compas-R | 10 | 5.0 | 4.3 | 3.8 | 3.9 | 4.4 | 4.1 | 13.7 | 13.0 |
| Gender | 100 | 3.3 | 2.7 | 2.6 | 2.3 | 2.7 | 2.8 | 8.0 | 7.4 |
| | 1000 | 1.4 | 2.1 | 1.3 | 1.3 | 1.4 | 3.0 | 1.8 | 2.4 |
| Compas-VR | 10 | 4.0 | 3.9 | 4.4 | 4.7 | 2.4 | 2.9 | 6.5 | 6.4 |
| Race | 100 | 3.1 | 2.8 | 3.4 | 3.3 | 2.0 | 2.1 | 3.7 | 3.6 |
| | 1000 | 0.8 | 1.5 | 0.8 | 0.8 | 0.8 | 2.5 | 0.9 | 1.8 |
| Compas-VR | 10 | 5.4 | 4.8 | 5.3 | 5.2 | 6.3 | 8.2 | 9.8 | 9.0 |
| Gender | 100 | 3.4 | 3.0 | 3.1 | 3.3 | 4.4 | 5.4 | 4.5 | 4.2 |
| | 1000 | 0.9 | 1.2 | 0.9 | 1.5 | 1.0 | 1.7 | 0.9 | 0.9 |
| Ricci | 10 | 14.6 | 14.2 | 7.9 | 8.1 | 2.1 | 2.0 | 1.6 | 2.1 |
| Race | 20 | 9.8 | 13.6 | 7.1 | 6.6 | 1.5 | 1.6 | 2.1 | 2.5 |
| | 30 | 6.5 | 12.1 | 4.6 | 4.2 | 1.1 | 1.4 | 2.0 | 2.3 |

## Appendix: Proof of Lemma 2.1

**Lemma 2.1.** *Given a prediction model $M$ and score distribution $P(s)$, let $f_g(s; \phi_g) : [0, 1] \to [0, 1]$ denote the calibration model for group $g$; let $f_g^*(s) : [0, 1] \to [0, 1]$ be the optimal calibration function which maps $s = P_M(\hat{y} = 1|g)$ to $P(y = 1|g)$; and $\Delta^*$ is the true value of the metric. Then the absolute value of expected estimation error w.r.t. $\phi$ can be bounded as: $|\mathbb{E}_\phi \Delta - \Delta^*| \leq \|\bar{f}_0 - f_0^*\|_1 + \|\bar{f}_1 - f_1^*\|_1$, where $\bar{f}_g(s) = \mathbb{E}_{\phi_g} f_g(s; \phi_g), \forall s \in [0, 1]$, and $\| \cdot \|_1$ is the expected $L^1$ distance w.r.t. $P(s|g)$.*

*Proof.*

$$
\begin{aligned}
|\mathbb{E}_\phi \Delta - \Delta^*| &= |(\mathbb{E}_{\phi_1}\theta_1 - \mathbb{E}_{\phi_0}\theta_0) - (\theta_1^* - \theta_0^*)| \\
&\leq |\mathbb{E}_{\phi_0}\theta_0 - \theta_0^*| + |\mathbb{E}_{\phi_1}\theta_1 - \theta_1^*| && \text{(triangle inequality)} \\
&= \|\bar{f}_0 - f_0^*\|_1 + \|\bar{f}_1 - f_1^*\|_1 && \text{(Lemma 2.2)}
\end{aligned}
$$

$\square$

**Lemma 2.2.** *Given a prediction model $M$ and score distribution $P(s)$, let $f_g(s; \phi_g) : [0, 1] \to [0, 1]$ denote the calibration model for group $g$; let $f_g^*(s) : [0, 1] \to [0, 1]$ be the optimal calibration function which maps $s = P_M(\hat{y} = 1|g)$ to $P(y = 1|g)$; and $\theta^*$ is the true value of the accuracy. Then the absolute value of expected estimation error w.r.t. $\phi$ can be bounded as: $|\mathbb{E}_\phi \theta_g - \theta_g^*| \leq \|\bar{f}_g - f_g^*\|_1$, where $\bar{f}_g(s) = \mathbb{E}_{\phi_g} f_g(s; \phi_g), \forall s \in [0, 1]$, and $\| \cdot \|_1$ is the expected $L^1$ distance w.r.t. $P(s|g)$.*

*Proof.*

$$
\begin{aligned}
\theta_g^* &= P(y = 0, \hat{y} = 0|g) + P(y = 1, \hat{y} = 1|g) \\
&= \int_{s<0.5} P(y = 0|s)P(s|g)ds + \int_{s>=0.5} P(y = 1|s)P(s|g)ds \\
&= \int_{s<0.5} (1 - f^*(s))P(s|g)ds + \int_{s>=0.5} f^*(s)P(s|g)ds
\end{aligned}
$$

Similarly, our method makes prediction about groupwise accuracy with calibrated scores given $P(\phi)$:

$$
\begin{aligned}
\mathbb{E}_{\phi_g}\theta_g &= \mathbb{E}_{\phi_g} \int_{s<0.5} (1 - f_g(s; \phi))P(s|g)ds + \int_{s\geq0.5} f_g(s; \phi)P(s|g)ds \\
&= \int_{s<0.5} (1 - \mathbb{E}_\phi f_g(s; \phi))P(s|g)ds + \int_{s>=0.5} \mathbb{E}_\phi f_g(s; \phi)P(s|g)ds \\
&= \int_{s<0.5} (1 - \bar{f}_g(s))P(s|g)ds + \int_{s>=0.5} \bar{f}_g(s)P(s|g)ds
\end{aligned}
$$

Then the absolute estimation bias of estimator $\mathbb{E}_{\phi \in \Phi}\theta_\phi$ is:

$$
\begin{aligned}
|\mathbb{E}_\phi \theta_g - \theta_g^*| &= |\int_{s<0.5} (\bar{f}(s) - f^*(s))P(s|g)ds + \int_{s>=0.5} (f^*(s) - \bar{f}(s))P(s|g)ds| \\
&\leq \int_{s<0.5} |\bar{f}(s) - f^*(s)|P(s|g)ds + \int_{s>=0.5} |f^*(s) - \bar{f}(s)|P(s|g)ds \\
&= \int_s |\bar{f}(s) - f^*(s)|P(s|g)ds \\
&= \|\bar{f} - f^*\|_1
\end{aligned}
$$

$\square$

## Footnotes

[1] https://github.com/algofairness/fairness-comparison/blob/master/fairness/preprocess.py

[2] https://archive.ics.uci.edu/ml/machine-learning-databases/adult

[3] http://archive.ics.uci.edu/ml/datasets/Bank+Marketing

[4] https://archive.ics.uci.edu/ml/machine-learning-databases/statlog/german

[5] https://github.com/propublica/compas-analysis

[6] https://github.com/propublica/compas-analysis

[7] https://ww2.amstat.org/publications/jse/v18n3/RicciData.csv

[8]"—" in Tables 2 and 3 there are entries where the frequentist estimates of TPR or FPR do not exist.