[Reviews · NeurIPS 2020]

Review 1

Summary and Contributions: This paper develops a Bayesian calibration model to estimate fairness outcomes in arbitrary models, with unlabelled data.

Strengths: - Clean and simple approach - Nice and clear empirical results - Addresses the case of unlabelled data

Weaknesses: - The novelty is somewhat minor, given existing work on Bayesian modelling of fairness - The paper focuses only on estimation and not on how to obtain fair policies - The process seems limited to fairness settings where Bayesian calibration is applicable

Correctness: Nothing significant

Clarity: Yes

Relation to Prior Work: Foulds et al, 2019. "Bayesian Modeling of Intersectional Fairness: The Variance of Bias" should be discussed

Reproducibility: Yes

Additional Feedback: This paper proposes to use Bayesian estimates of fairness metrics. It combines this with Bayesian calibration models (one for each protected attribute value in this particular case) in order to use unlabelled data. In light of existing work (Foulds et al 2019) on Bayesian modelling of fairness, the contribution is rather minor and is limited to the case where we have unlabelled data. The approach the authors use, as it is based on calibration, seems limited to rather specific notions of fairness where Bayesian calibration can be usefully applied. Although in l.64 the definition of calibration is correct, in l. 105-107 you write that $s_j = P_M(y_j = 1 | s_j)$. Since $j$ is a specific example, there should not be any randomness here. Calibration is a property of the classifier with respect to a data distribution, not a specific example. It would also be nice to discuss how you would extend this idea to other fairness metrics such as (the unfortunately named) calibration and balance. The paper would be significantly strengthened by generalising the approach to other settings where unlabelled data is available, and by using the Bayesian estimates to obtain fair(er) policies. I appreciate the author's response. Since the method is applicable to essentially any fairness metric classification, and since they have clarified the relation to prior work, I am revising my score. I wouldn't mind the paper being accepted, as the significance of a paper is a somewhat subjective criterion, but there is definitely nothing wrong with it and it's a valueable contribution in itself.


Review 2

Summary and Contributions: This paper proposes a Bayesian approach to reducing uncertainty in group fairness metrics in contexts where the amount of unlabeled examples greatly exceeds the number of labeled examples. The method uses the labeled data to infer the posterior densities of the parameters of per-group calibration curves, then marginalizes over these parameters to estimate the probabilities of different labeling outcomes for the unlabeled examples, before finally combining these into a posterior estimate of the desired fairness metric. The approach is well validated and shows significant performance gains over the frequentist and beta-binomial Bayesian estimates in the case of few labeled examples.

Strengths: The problem of uncertainty in fairness metrics is under-studied, despite being of extreme importance in many low-data situations where fairness is a concern (for example, predicting rare diseases or behaviors). This paper presents a nice simple approach to quantifying and reducing the uncertainty in fairness metrics, and is applicable to many use-cases and metrics.

Weaknesses: This paper acknowledges some limitations that I don't find consequential enough to prevent publication, for example: - the possibility that the CIs for their method might be overconfident if there are many labeled examples. - the challenge of balancing the bias-variance tradeoff for this method.

Correctness: Yes, the empirical methodology appears sound and well-justified.

Clarity: Yes, the paper is exceptionally clear and well written.

Relation to Prior Work: Yes, prior work in Bayesian fairness and data augmentation approaches are covered appropriately.

Reproducibility: Yes

Additional Feedback: I'm not sure it's appropriate to use "Eskimo" in Fig 2. My understanding is that that name is not the people's preferred name. UPDATE: The authors' feedback has not changed my evaluation of this good paper.


Review 3

Summary and Contributions: The paper addresses a timely topic.

Strengths: The empirical results seem promising.

Weaknesses: The hyper parameter selection (lines 152-153) needs more explanations. The values seem informative and it would be useful to see a proper sensitivity analysis. Also, it would be good to perform a comparison to related work (non-hierarchical variant) to assess contributions of the proposed variant. Eq 1 is central to the paper and the 'For example,' undermines the contributions. What other alternatives could be used or why to the particular form? More theoretical results would be useful to establish the relevance of the proposed approach. Posterior computations contain no novelties or contributions. --- Thanks for the rebuttal. Based on the feedback I am revising my score.

Correctness: The method requires more explanations.

Clarity: The paper is well written.

Relation to Prior Work: The related work should be better separated from the contributions of the paper.

Reproducibility: Yes

Additional Feedback:

[Author Response · NeurIPS 2020]

Thank you for your reviews! Your comments will be useful in revising our paper. We appreciate the positive comments, e.g., that our work addresses an "under-studied"/"timely" problem "of extreme importance," the method is "nice"/"simple"/"clean"/"well validated," the paper is "(exceptionally) clear"/"well written," results are "solid"/"significant"/"clear," with "significant performance gains."

We briefly summarize the different *major* concerns of the 3 reviewers before we respond to them separately. We note that **there were no shared weaknesses pointed out by the 3 reviewers.** The main concern of @R1 (overall score=4) is about the novelty and contribution of the paper, especially in the context of a specific recent line of work by Foulds et al. The reviewer also viewed focusing on assessment of fairness instead of learning fair models as a weakness. @R2 (overall score=7) characterizes our work as being a useful approach to quantifying and reducing uncertainty in fairness metrics, with broad applicability and significant performance gains, and "does not think the weaknesses of our work is consequential enough to prevent publication". @R4 (overall score=4) identified some potential weaknesses (prior sensitivity, desirability of more theoretical results) but did not identify major issues with the paper.

**@R1: "in light of Foulds et al 2019, the contribution is rather minor"** Thank you for pointing out this paper. We will certainly cite and discuss it in our revised paper since it shares a common starting point with our work of using Bayesian methods to assess fairness. *However, we disagree that our contribution is minor in light of this paper. The two papers complement each other: they address different problems and take different technical approaches*. We believe our work is substantially different from [1] in objective, methods and results and we ask Reviewer 1 to reconsider our paper in this light. **Objective:**, our work specifically focuses on the problem of *leveraging unlabeled data to generate better estimates of fairness metrics* given limited labeled data; in contrast, [1] focuses on assessing *intersectional fairness* when the amount of labeled data is extremely small and unlabeled data is unavailable. We also make a point in our work of emphasizing that estimates of fairness metrics can suffer from high variance even in the presence of relatively large amounts of labeled data (see Fig 1). **Methods**: We use Bayesian methods to calibrate scores for unlabeled datapoints for improved estimation of group fairness metrics, whereas [1] uses Bayesian methods to provide parametric smoothing among groups for improved estimation of intersectional fairness metrics based on labeled data. **Results**: We evaluate our methods across a broad range of different datasets and different models (mechanisms) and demonstrate substantial improvements in accuracy of estimates of multiple group fairness metrics, whereas [1]'s results are focused mainly about the accuracy ratio between intersectional groups on two (semi-synthetic) datasets.

**@R1: "focuses only on estimation and not on how to obtain fair policies"**. We agree that obtaining fair policies is an important problem, but we also believe that for a given model, trained fairly or not, *independent and accurate assessment of its fairness is important and under-studied* (also the motivation in [1]), particularly when users only have access to a blackbox predictor. We will make this point clearer in revision. **@R1: "seems limited to fairness settings where Bayesian calibration is applicable"**: We emphasize that our approach is applicable to any classification setting and we **do not need** any special setup to apply our method. The only requirement is that there is both labeled and unlabeled data available from the deployment environment. @R1: notation in l.105-107: We agree this notation is confusing and will remove this equation (its not needed). **@R1: "discuss how you would extend this idea to other fairness metrics."** Good point, we agree. We can directly extend our approach to handle metrics such as calibration and balance as well as ratio-based metrics and we will this discussion of such extensions to the paper. All fairness metrics which are defined as deterministic functions of model score $S$, label $Y$ and sensitive attribute $A$ (for concreteness we demonstrated with 3 popular fairness metrics in the paper) can be approximated on unlabeled data with our proposed method.

**@R2: "the possibility that the CIs might be overconfident if there are many labeled examples."** Good point. On page 13 of Appendix, we empirically validated that our method provides reasonably well-calibrated CIs, but we believe there is room for further improvement in this area. **@R2: "the challenge of balancing the bias-variance tradeoff for this method."** We agree that this is an interesting direction for future work. We provide some theoretical considerations in lines 165-177 and also acknowledge this issue in a brief discussion in lines 242-250: but there is certainly room for more work on this front. **@R2:Eskimo -> Inuit**: Thank you for spotting this! We will update it in the paper.

**@R4: "proper sensitivity analysis"**: We agree that systematic sensitivity analysis is lacking and we will add it to the Supplement. The values for the priors were selected based on consideration on knowledge of ranges of miscalibration one typically sees with trained classifiers—and we also found that the same priors worked well across a large range of datasets and models without any need for tuning (see also lines 153-156). **@R4: "non-hierarchical variance**: Thanks for suggesting the ablation study, we will add it to the Supplement. **@R4: Equation 1**: The phrase "For example" was loosely worded and we will remove it in the revision to avoid potential confusion. **@R4: "more theoretical results to establish the relevance"**: We agree that more theoretical results are important going forward—please see our response to R2 on this point. **@R4: "posterior computations contain no novelties or contributions"**: We use standard MCMC for posterior computation and it works well in our experiments in terms of both accuracy and runtime.

[Meta-Review · NeurIPS 2020]

This paper focuses on the problem of leveraging unlabelled data to generate better estimates of fairness metrics given limited labelled data. All three reviewers agree that the manuscript makes a valuable contribution and is conceptually and mathematically sound. The significance of the contribution (an auditor tool only, instead of an auditor plus a mitigation tool) is however at the low side.